# LAST LAYER LOGITS TO LOGIC: EMPOWERING LLMS WITH LOGIC-CONSISTENT STRUCTURED KNOWLEDGE REASONING

## ABSTRACT

Large Language Models (LLMs) achieve excellent performance in natural language reasoning tasks through pre-training on vast unstructured text, enabling them to understand the logic in natural language and generate logic-consistent responses. However, the representational differences between unstructured and structured knowledge make LLMs inherently struggle to maintain logic consistency, leading to *Logic Drift* challenges in structured knowledge reasoning tasks such as Knowledge Graph Question Answering (KGQA). Existing methods address this limitation by designing complex workflows embedded in prompts to guide LLM reasoning. Nevertheless, these approaches only provide input-level guidance and fail to fundamentally address the *Logic Drift* in LLM outputs. Additionally, their inflexible reasoning workflows cannot adapt to different tasks and knowledge graphs. To enhance LLMs' logic consistency in structured knowledge reasoning, we specifically target the logits output from the autoregressive generation process. We propose the *Logits-to-Logic* framework, which incorporates logits strengthening and logits filtering as core modules to correct logical defects in LLM outputs. Extensive experiments show that our approach significantly improves LLMs' logic consistency in structured knowledge reasoning and achieves state-of-the-art performance on multiple KGQA benchmarks.

## 1 INTRODUCTION

Large language models (LLMs) (Achiam et al., 2023; Brown et al., 2020a; Chowdhery et al., 2023) are pre-trained on unstructured natural-language corpora (Rawte et al., 2023), granting them strong logical reasoning abilities. They can easily follow text-level logic and achieve excellent performance on natural-language inference tasks (Wei et al., 2022; Khot et al., 2022).

Structured knowledge exists widely in the real world, particularly playing an important role in reasoning systems by providing precise and explainable paths (Ji et al., 2022; Sun et al., 2018). Knowledge graphs (KGs) (Auer et al., 2007; Bollacker et al., 2008a) represent a crucial form of structured knowledge. Research on structured knowledge reasoning primarily focuses on Knowledge Graph Question Answering (KGQA), which involves answering natural language questions based on structured factual information stored in KGs (Miller et al., 2016). As a result, leveraging LLMs for KGQA has attracted considerable attention (Yasunaga et al., 2021). However, despite significant achievements, LLMs still face challenges in such structured knowledge reasoning (Ji et al., 2023). Unlike natural language text, structured KGs are constrained by predefined schemas (Suchanek et al., 2007) and represented in triplet format (*head entity, relation, tail entity*) (Krompaß et al., 2015). This representational difference makes it inherently difficult for LLMs to fully understand structured knowledge and maintain logic consistency, leading to Logic Drift challenges in structured knowledge reasoning tasks such as KGQA. Fig. 2 illustrates two main forms of Logic Drift: (1) LLMs often output reasoning paths that do not exist in the KG; (2) LLMs generate reasoning paths that are semantically irrelevant to the question logic. It is important to note

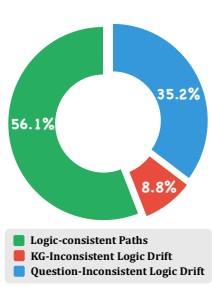

Figure 1: Logic Drift ratio statistics of ToG, DoG, GCR, and KG-CoT on 100 sampled instances from the CWQ dataset.

that Logic Drift is a specific type of hallucination in KGQA, where LLMs generate reasoning paths that are logically inconsistent with question intent or KG structure, distinct from general hallucination involving fabricated facts (Detailed discussion in Appendix L). To understand the underlying mechanism of Logic Drift, we analyze LLMs' internal representations. We observe from LLMs' last layer logits distribution that tokens in reasoning paths semantically irrelevant to question logic (Orange highlighted) and non-existent in KG (Gray highlighted) have high logits values. To mitigate this phenomenon, *we can map the logic of questions and KG into distributions in logits probability space. We highlight that the above **Logic Drift issues stem from the inconsistency between LLMs' output distributions and the logical distributions of the structured KG and question***.

Existing work mainly addresses Logic Drift by designing complex agent-based frameworks. ToG (Sun et al., 2023) adopts a step-by-step procedure, exploring entities and relations one hop at a time to strengthen the model's grasp of structured logic. DoG (Ma et al., 2025) designs three agent roles (simplify, critic, linguist) to iteratively decompose questions and correct reasoning logic through single-step modifications. KG-CoT (Zhao et al., 2024) and GCR (Luo et al., 2025) propose large-small model collaboration paradigms, using lightweight agents to pre-filter candidate paths that are logically aligned with the question before LLMs reasoning. In summary, **most advanced methods embeds complex, task-specific workflows in prompts, offering only input-level guidance that neither resolves Logic Drift fundamentally in the output (shown in Fig. 1) nor adapts well to diverse tasks and structured KGs**.

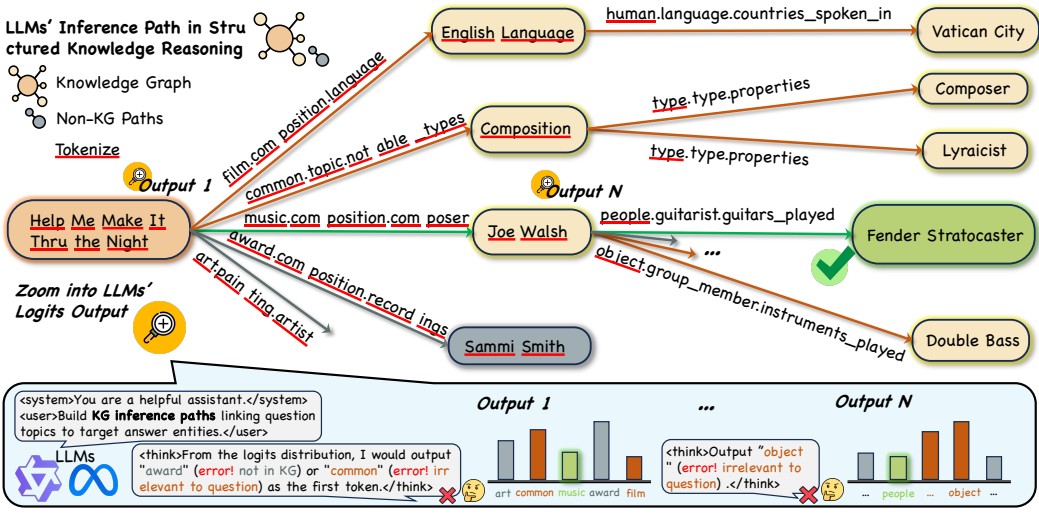

Figure 2: Logic Drift in LLM's output. Orange highlights reasoning paths/tokens semantically irrelevant to the question's logic. Gray highlights reasoning paths/tokens inconsistent with structured KG logic (i.e., hallucinated paths that LLMs output but don't exist in the KG).

To overcome their limitation, we target LLMs' output process directly. LLMs generate the next token based on logits distribution at the last layer, which serves as the critical decision-making step. Based on this, we propose the Logits-to-Logic framework acting directly on the last-layer logits to improve logic consistency in structured knowledge reasoning at its source. Logits-to-Logic proceeds in three stages: **(1) Logic compiling**: We compile legal paths in the KG into an NFA (Non-deterministic Finite Automaton) and score NFA paths. This prepares for aligning LLMs' output distribution with the logical distributions of the KG and question; **(2) Logits strengthening**: We enhance logits values that align with the question's semantic logic in the NFA using differentiation and scaling techniques; **(3) Logits filtering**: We use legal paths in the NFA to constrain the logits of illegal tokens.

Our major contributions are as follows:

- We highlight that the key challenge of LLMs in structured knowledge reasoning lies in the inconsistency between their outputs and the logical distributions of KG and question. Unlike previous work focusing solely on input, we propose addressing logic drift from the output perspective.

- We propose the Logits-to-Logic framework and design logits strengthening and logits filtering modules to fundamentally address the Logic Drift from the output perspective.

- Extensive experimental results on multiple KGQA benchmarks demonstrate that our method significantly improves LLMs' logic consistency in structured knowledge reasoning and achieves state-of-the-art performance, while being directly transferable to different KGs and tasks.

## 2 RELATED WORK

**Logical LLMs Reasoning.** LLMs often exhibit logical inconsistencies like hallucinations and semantic errors during complex reasoning. Existing work mainly addresses this through designing intricate reasoning chains or frameworks. Chain-of-Thought (CoT) (Wei et al., 2022) guides models through intermediate steps, breaking complex problems into sub-problems. Graph-of-Thought (GoT) (Yao et al., 2023a) and Tree-of-Thought (ToT) (Besta et al., 2024) extend this into graph and tree structures for exploring multiple reasoning paths. These methods use intricate prompts to guide LLMs toward logic-consistent reasoning, reducing logic drift in reasoning chains. ReACT (Yao et al., 2023b) employs "Thought-Act-Observe" workflows allowing LLMs to obtain real-time feedback for correcting reasoning logic. Reflexion (Shinn et al., 2023) uses iterative "Act-Eval-Reflection" workflows to improve reasoning performance. However, these methods rely on input-level guidance and operate within natural language space, still facing logic-inconsistency challenges in structured knowledge reasoning tasks requiring strict schema constraints.

**Agentic Structured Knowledge Reasoning.** Advanced KGQA methods treat structured KGs as dynamic environments and design complex agent-based frameworks to maintain logic-consistency during LLM reasoning . ToG (Sun et al., 2023), PoG (Chen et al., 2024) designs step-by-step reasoning processes, enhancing LLMs' understanding of structured knowledge logic through single-step entity and relation exploration on KGs. DoG (Ma et al., 2025) designs three agent roles (simplify, critic, linguist) to iteratively decompose questions and correct reasoning logic through single-step modifications. Similarly, KG-Agent (Jiang et al., 2025) and SymAgent (Liu et al., 2025) design planner, toolbox, and executor roles for automated reasoning, but their custom toolboxes cannot cover all logical patterns in KGs. KG-CoT (Zhao et al., 2024) and GCR (Luo et al., 2025) propose large-small model collaboration paradigms, using smaller-parameter agents to pre-filter candidate paths that align with question semantic logic before LLM reasoning. DARA (Fang et al., 2024) and GoG (Xu et al., 2024) embed the "Thought-Act-Observe" workflow into structured knowledge reasoning, allowing LLMs to self-correct reasoning logic through real-time feedback. While these methods improve KGQA performance through multi-step reasoning and diverse agent roles, they still operate in natural language space and fail to fundamentally solve logic-inconsistent outputs. Moreover, their rigid workflows lack flexibility for different reasoning tasks and KGs. Therefore, we propose operating at the logits distribution level of LLM outputs, fundamentally correcting LLM reasoning by mapping KG and question logical distributions into logits probability space to ensure logic-consistency. A detailed introduction to existing KGQA work can be found in the Appendix A.

## 3 METHODS

### 3.1 PROBLEM DEFINITION

**Knowledge Graph.** Given a knowledge graph $G$ as a collection of structured knowledge, it's organized in triplet format: $G = \{(e_s, r, e_o) \in E \times R \times E\}$, where $E$ is the entity set and $R$ is the relation set. Multiple consecutive triplets with matching head-tail entities can form paths in $G$, which provide precise and explainable reasoning forms for reasoning models $M_\theta$. We define a KG path as: $s = e_1 \rightarrow r_1 \rightarrow e_2 \rightarrow r_2 \rightarrow e_3 \rightarrow ... \rightarrow r_{l-1} \rightarrow e_l, \forall e_i \in E, r_i \in R$.

**Knowledge Graph Question Answering.** In KGQA tasks, for a given question $q$, we can extract topic entities $e^{topic} = \left\{ e_i^{topic} \in E \right\}$ from it. The topic entities typically serve as the starting point for reasoning in $G$. Our goal is to enable LLM with parameters $\theta$ ($M_\theta$) to perform structured knowledge reasoning on $G$ and find answer paths as follows: $s_+^{e^{topic}} = \left\{ e^{topic} \rightarrow r_1 \rightarrow e_2 \rightarrow r_2 \rightarrow e_3 \rightarrow ... \rightarrow r_{l-1} \rightarrow a \,\middle|\, a \in E \right\}$, where $a$ is the answer entity. We de-

fine the logical distributions of LLMs' original output, question and KG as $\mathcal{D}_\theta, \mathcal{D}_q, \mathcal{D}_G$. Our goal is to enable LLMs to perform logic-consistent reasoning, which helps derive $s_+^{e^{topic}}$:

$$\left\{ s_+^{e^{topic}} \right\} \propto \mathcal{D}_{q,G} \sim \underset{\mathcal{D}_\theta}{argmax} P_\theta \left( a | q, G \right)$$

**State Tranfer Reasoning.** LLMs $M_\theta$ generate logits distribution for the next token through autoregressive generation, with current output depending on the previous sequence. We can simulate this logits output process using state transition models. Reasoning paths in KGs represent entity-to-entity symbolic transitions, which naturally fit state transitions. Therefore, LLMs' reasoning process and structured KGs are naturally suitable for modeling as Non-deterministic Finite Automaton (NFA), providing a feasible approach for aligning LLMs' outputs with structured knowledge distributions to solve Logic Drift. NFA allows transitions from one state to multiple possible states through the same input. This non-deterministic feature matches LLMs' probabilistic mechanism that generates multiple possible tokens at each step, providing an ideal framework for mapping LLMs' token generation to KG entity transition paths. We can model the structured KGs and LLMs autoregressive generation as: $NFA = \left( S_{0:end}, \Sigma, \delta, e^{topic}, S \right)$, where $S$ represents accepting states (i.e., all legal reasoning paths in $G$), and $S_{0:end}$ represents all legal reasoning state sets. $\Sigma$ is the LLM's vocabulary, i.e., the set of all tokens $T = \{ t \in \Sigma \}$. $\delta$ denotes the transition function: $\delta(t) = t \times S_{i:end} \to S_{i+1:end}, t \in \Sigma$.

## 3.2 LOGITS-TO-LOGIC FRAMEWORK

**Question:** What kind of guitar was used by the lyricist for *Help Me Make It Thru the Night?*

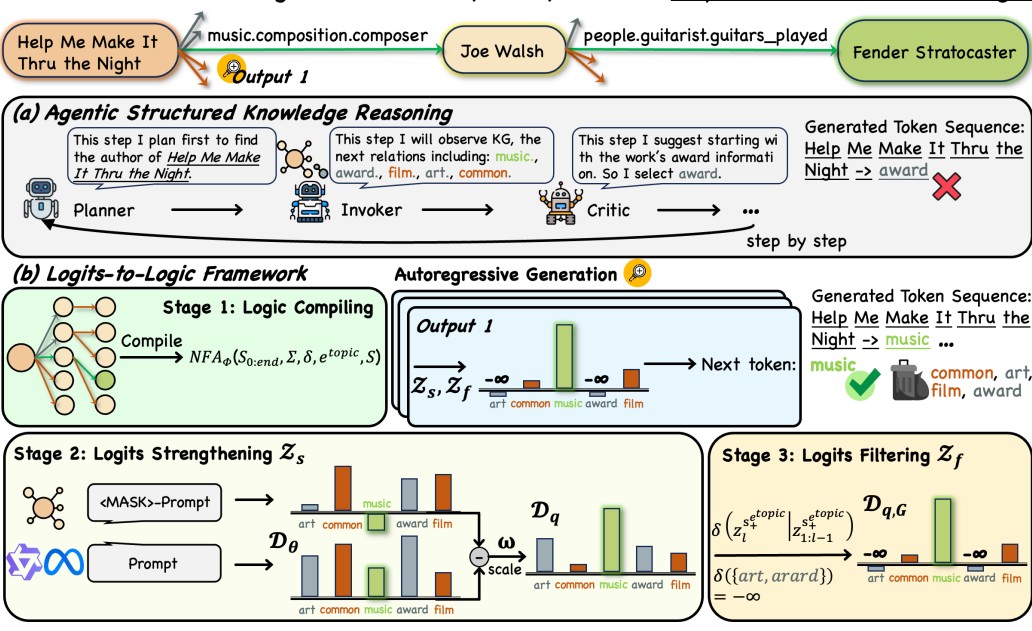

Figure 3: (a) Previous agentic methods attempt to maintain logic consistency by designing complex workflows or prompt engineering to guide LLMs from the input level; (b) Overview of our framework: we align LLMs' last-layer logits distribution with question and KG logic through Logits Strengthening ($\mathcal{Z}_s$) and Filtering ($\mathcal{Z}_f$).

As shown in Fig. 3, given question $q$ and $G$, Logits-to-Logic includes 3 stages during reasoning: **(1) Logic Compiling**: We compile all legal paths in the KG into NFA. Based on this, we use sentence-transformer to score legal paths $S$ in the NFA with $q$, where scores indicate reasoning paths similar to question semantic logic. This prepares for aligning LLMs' outputs with the logical distributions of KG and question; **(2) Logits Strengthening $\mathcal{Z}_s$**: We enhance logits of high-scoring legal paths in the NFA through differentiation and scaling, making the logits distribution closer to question logic to address the inconsistency between LLMs' outputs and question logic; **(3) Logits Filtering $\mathcal{Z}_f$**: We

constrain logits values corresponding to tokens that do not belong to legal paths in the NFA, thereby aligning LLMs' outputs with the logical distribution of structured KG.

In general, our reasoning objective during decoding is: using $\mathcal{Z}_s$ and $\mathcal{Z}_f$ to align LLMs' logits with the logical distributions of $q$ and $G$ ($s_+^{e^{topic}} \cup s_-^{e^{topic}}$), making LLMs output correct answer paths $s_+^{e^{topic}}$ while avoiding incorrect paths $s_-^{e^{topic}}$.

$$P_\theta\left(a|q,G\right) \propto P_{\theta,q,G}\left(a|q,G\right) = P_{\theta,q,G}\left(a\Big|q, \left\{s_+^{e^{topic}}\right\}, \left\{s_-^{e^{topic}}\right\}\right)$$

$$\propto \mathcal{D}_{q,G} \sim P_\theta\left(a\Big|q, \left\{s_+^{e^{topic}}\right\}\right) \cdot \underbrace{P_{\theta,q}\left(\left\{s_+^{e^{topic}}\right\}\Big|q,G\right)}_{\text{Question Logical Distribution } \mathcal{D}_q} \cdot \underbrace{P_{\theta,G}\left(\left\{s_+^{e^{topic}}\right\}\Big|q,G\right)}_{\text{KG Logical Distribution } \mathcal{D}_G}$$

### 3.2.1 LOGIC COMPILING

As described in Sec. 3.1, LLMs Structured Knowledge Reasoning naturally aligns with NFA. Therefore, in this stage, we compile the logic of structured KG and question into NFA. We set the NFA initial state as the question's topic entity $e^{topic}$, use LLMs' vocabulary $\Sigma$ as input, and compile all legal KG paths $S$ into NFA. For example, for the question *What kind of guitar was used by the lyricist for Help Me Make It Thru the Night?*, the initial state is *Help Me Make It Thru the Night*. We can decompose the accepting states (i.e., legal reasoning path in $G$) *Help Me Make It Thru the Night → common.topic.notable_types → Composition → type.type.properties → Lyricist* into multiple legal states. Legal states refer to subsequences of tokens from accepting states (e.g., *Help Me Make It Thru the Night → common.topic.notable*). We determine acceptable tokens for each state to obtain the transition function $\delta$. For example, from the state *Help Me Make It Thru the Night → common.topic.notable*, $\delta$ would specify that the next acceptable token is *_types*. We then add question semantic logic into NFA using sentence-transformer $M_\Phi$ to score semantic similarity between question and paths, obtaining high-scoring paths (Details in Appendix G, H). Therefore, we obtain an NFA considering both KG and question logical distributions: $NFA_\Phi = \left(S_{0:end}, \Sigma, \delta, e^{topic}, S\right)$.

### 3.2.2 LOGITS STRENGTHENING

As shown in Fig. 3 (a), to enable LLMs to perform reasoning consistent with question logic, previous work designs complex agent collaboration but still fails. Observing LLMs' raw logits distribution, we find tokens irrelevant to the question have high logits values (Orange-highlighted *common* and *film* in Fig. 3), while correct reasoning path tokens have low logits values (Green-highlighted *music*). To address this inconsistency between LLMs outputs and question logic, we need to strengthen correct token logits influence.

Contrastive Decoding (Li et al., 2023) operates on expert and amateur model logits to enhance correct tokens while excluding irrelevant ones. Inspired by this, we design logits strengthening for structured KGs to enhance logits values of tokens that align with question semantic logic. We use $P_\theta\left(\left\{s_+^{e^{topic}}\right\}\Big|q,G\right)$ to represent the original probability of LLMs outputting answer paths, with $\mathcal{Z}_s$ aiming to boost the probability of $s_+^{e^{topic}}$ in the output distribution $P_{\theta,q}\left(\left\{s_+^{e^{topic}}\right\}\Big|q,G\right)$. We treat high-scoring NFA paths from stage 1 as answer paths, while low-scoring paths are noise paths unrelated to the question. To strengthen answer path logits influence, we design two prompts: the original prompt and a masked version that replaces answer paths with special MASK tokens (Details in Appendix G, H). The masked output amplifies noise path influence and deviates from question logic. We calculate the difference between original and masked outputs, then multiply by coefficient $\omega$ to amplify correct token logits, then add the result to the masked outputs' logical distribution.

$$\mathcal{D}_q \sim P_{\theta,q}\left(\left\{s_+^{e^{topic}}\right\}\Big|q,G\right)$$

$$= \underbrace{\omega \cdot P_\theta\left(\left\{s_+^{e^{topic}}\right\}\Big|q, \left\{s_+^{e^{topic}}\right\}, \left\{s_-^{e^{topic}}\right\}\right) + (1-\omega) \cdot P_\theta\left(\left\{s_+^{e^{topic}}\right\}\Big|q, \text{MASK}, \left\{s_-^{e^{topic}}\right\}\right)}_{\text{Logits Strengthening } \mathcal{Z}_s}$$

We explored optimal $\omega$ values from -1 to 10 in Fig. 4 Right. Through logits strengthening, logits values of tokens in answer paths consistent with question logic are enhanced.

### 3.2.3 Logits Filtering

The logits distribution from $\mathcal{Z}_s$ considers question semantic logic, but still has high-value logits for illegal tokens not in the KG (Gray-highlighted *art* and *award*). Sampling these illegal tokens causes reasoning chain breaks. To address logic-inconsistency between LLMs outputs and structured KG, we use logits filtering $\mathcal{Z}_s$ to constrain illegal token generation. Specifically, we use NFA's transition function $\delta$ to guide LLMs in generating legal paths.

$$
P_\theta\left(a|q,G\right) \propto P_\theta\left(a\Big|q,\left\{s_+^{e^{topic}}\right\}\right) \cdot \mathcal{D}_q \cdot \prod_{l=1}^{|\{s_+^{e^{topic}}\}|} P_\theta\left(t_l^{s_+^{e^{topic}}}\Big|q,t_{1:l-1}^{s_+^{e^{topic}}}\right)\delta\left(t_l^{s_+^{e^{topic}}}\Big|t_{1:l-1}^{s_+^{e^{topic}}}\right)
$$

$$
\propto P_\theta\left(a\Big|q,\left\{s_+^{e^{topic}}\right\}\right) \cdot \mathcal{D}_q \cdot \underbrace{\prod_{l=1}^{|\{s_+^{e^{topic}}\}|} z_l^{s_+^{e^{topic}}} \cdot \delta\left(t_l^{s_+^{e^{topic}}}\Big|t_{1:l-1}^{s_+^{e^{topic}}}\right)}_{\text{Logits Filtering } \mathcal{Z}_f},
$$

where $z$ represents the logit value corresponding to token $t$. For example, when encountering illegal tokens *art* and *award*, we set $\delta\left(\{art, award\}\right) = -\infty$ to constrain their generation.

In summary, we use $\mathcal{Z}_s$, $\mathcal{Z}_f$ to align LLMs outputs with the logical distributions of KG and question, guiding LLMs to perform logic-consistent reasoning in structured knowledge.

$$
P_\theta\left(a|q,G\right) \propto \mathcal{D}_{q,G} \sim P_\theta\left(a\Big|q,\left\{s_+^{e^{topic}}\right\}\right) \cdot \left(\mathcal{D}_q\mathcal{D}_G\right)
$$

## 4 Experiments

In this section, we present our experimental settings and evaluation results with detailed analysis. In this section, we aim to explore the following key research questions (**RQ**) in our work. **RQ1**: Can Logits-to-Logic outperform baselines and achieve state-of-the-art performance across multiple KG reasoning benchmarks? **RQ2**: How do logits strengthening $\mathcal{Z}_s$ and logits filtering $\mathcal{Z}_f$ help LLMs maintain logic-consistency in structured knowledge reasoning? **RQ3**: Can Logits-to-Logic easily transfer to different KGs and tasks while maintaining robustness and flexibility? **RQ4**: How does Logits-to-Logic perform on different backbones of current mainstream open-source LLMs? **RQ5**: Can Logits-to-Logic effectively reduce logic drift problems in structured knowledge reasoning?

### 4.1 Experimental Settings

**Datasets and Tasks.** We select multiple KG reasoning benchmarks covering different structured reasoning subtasks and KGs. We use Freebase-based (Bollacker et al., 2008b) **CWQ** (Talmor & Berant, 2018), **WebQSP** (Yih et al., 2016), **GrailQA** (Gu et al., 2021), and **Simple Questions (SQ)**, as well as larger Wikidata-based (Vrandečić & Krötzsch, 2014) **QALD10-en** (Perevalov et al., 2022), **T-REx** (Elsahar et al., 2018), and **Zero-shot RE** (Petroni et al., 2020). CWQ, WebQSP, GrailQA, and QALD10-en are multi-hop complex reasoning datasets, Simple Questions is a single-hop reasoning dataset, while T-REx and Zero-shot RE are slot filling datasets. This dataset and task setup comprehensively validates the method's capability and robustness. Detailed dataset and task information is in the Appendix B.

**Baselines.** We select two mainstream KG reasoning methods as baselines: **(1) LLMs Reasoning** methods use prompt engineering with LLMs for structured knowledge reasoning; **(2) Agentic Reasoning** methods treat KGs as dynamic environments and design intricate prompts and workflows to guide multi-agent collaborative reasoning. Detailed information about these methods is in the Appendix E.

**Evaluation Metrics.** We use Hit@1 and F1 as evaluation metrics. Hit@1 considers whether the correct answer exists in the model's top-ranked prediction, while F1 considers both prediction accuracy and answer coverage.

**Implementation Details.** We use LLaMA-3.1-8b as the default LLM with beam search decoding and default beam size of 20. We set the default logits strengthening value $\omega$ to 2.0 and use the lightweight 22M-parameter sentence-transformer-all-MiniLM-L6-v2 (Reimers & Gurevych, 2019) as the NFA legal path scoring model. Before testing, we perform SFT (supervised fine-tuning) to teach the model correct path output format using 1/10 randomly sampled data from CWQ and WebQSP training sets. We implement our method in PyTorch on Ubuntu 20.04.1 LTS servers with two A800 GPUs. More details are in the Appendix D.

## 4.2 MAIN RESULTS (RQ1)

Tab. 1 shows the comparison results between Logits-to-Logic and advanced methods. We select methods from both LLMs Reasoning and Agentic Reasoning paradigms as baselines. Results show that our method achieves state-of-the-art performance on three multi-hop complex QA datasets (WebQSP, CWQ, GrailQA) and one single-hop simple QA dataset (Simple Questions; SQ). Due to LLMs' inherent logic drift defects in structured knowledge reasoning, the LLMs Reasoning paradigm performs significantly worse than agent-based paradigms.

Among agent-based methods, our approach outperforms RoG and GoG by 9.7% and 11% on WebQSP, surpasses KG-Agent, SymAgent, and GCR by 8.6%, 22%, and 5% on CWQ, exceeds DoG, PoG, and DARA by 2%, 5.5%, and 5% on GrailQA, and beats KG-CoT and ToG by 2.5% and 13.6% on Simple Questions. These methods use large models like ChatGPT and GPT4, while our method only requires the smaller LLaMA-3.1-8b model, demonstrating our method's superior performance on structured knowledge reasoning tasks.

Table 1: Hit@1 Performance comparison of Logits-to-Logic and various baselines on three multi-hop and one single-hop KGQA datasets, with the **best results in bold**. *, §, †, ‡ indicates w/ ChatGPT, GPT4, LLaMA2-13b, LLaMA3.1-8b, respectively.

| Method | Muti-hop | | | Single-hop |
|---|---|---|---|---|
| | WebQSP | CWQ | GrailQA | SQ |
| *LLMs Reasoning* | | | | |
| IO prompt* | 63.3 | 37.6 | 29.4 | 20.0 |
| CoT* | 62.2 | 38.8 | 28.1 | 20.3 |
| SC* | 61.1 | 45.4 | 29.6 | 18.9 |
| *Agentic Reasoning* | | | | |
| StructGPT | 72.6 | 54.3 | - | - |
| KD-CoT | 73.7 | 50.5 | - | - |
| KG-CoT* | 84.9 | 62.3 | - | 77.8 |
| RoG | 85.7 | 62.6 | - | - |
| DoG | 91.0 | 56.0 | 80.0 | - |
| ToG-R* | 75.8 | 58.9 | 56.4 | 45.4 |
| ToG* | 76.2 | 57.1 | 68.7 | 53.6 |
| ToG-R§ | 81.9 | 69.5 | 80.3 | 58.6 |
| ToG§ | 82.6 | 67.6 | 81.4 | 66.7 |
| GCR‡ | 92.2 | 75.8 | - | - |
| PoG* | 82.0 | 63.2 | 76.5 | - |
| GoG | 84.4 | 75.2 | - | - |
| DARA† | 30.3 | - | 77.0 | - |
| KG-Agent | 83.3 | 72.2 | - | - |
| SymAgent | 78.5 | 58.8 | - | - |
| **Ours‡** | **95.4** | **80.8** | **82.0** | **80.3** |

## 4.3 LOGIC-CONSISTENT ANALYSIS (RQ2)

We conduct module ablation studies to verify the effectiveness of logits strengthening $\mathcal{Z}_s$ and logits filtering $\mathcal{Z}_f$. As shown in Tab. 2, we test three settings: (1) w/o $\mathcal{Z}_s$; (2) w/o $\mathcal{Z}_f$; (3) w/o $\mathcal{Z}_s$ & $\mathcal{Z}_f$. Results show that removing $\mathcal{Z}_s$ decreases performance by 1.5% and 6.9% on WebQSP and CWQ respectively, indicating that LLMs outputs cannot align with question logic, leading to more reasoning errors. Removing $\mathcal{Z}_f$ prevents LLMs outputs from aligning with structured KG logical distribution, causing performance drops of 9.2% and 1.4% on WebQSP and CWQ respectively. Removing both modules leads to significant performance degradation. This verifies that logits strengthening & filtering help LLMs maintain logic-consistent reasoning in structured knowledge and effectively improve reasoning performance.

We explore the impact of strength value $\omega$ in the logits strengthening module. Fig. 4 **Right** shows performance changes with different strength values. We investigate the effects of weakening ($\omega = -1.0$) and different degrees of strengthening ($\omega = 1 \sim 10$). Results show that when $\omega = -1.0$, weakening correct token logits influence causes LLMs outputs to deviate from question logic, leading to performance decline. When $\omega > 3.0$, performance drops to varying degrees. We find that excessive $\omega$ values over-amplify the logits influence of answer path tokens in LLMs

Table 2: Ablation study of Logits-to-Logic's core modules, comparing the results after removing the $\mathcal{Z}_f$ and $\mathcal{Z}_s$ modules separately.

| Method | WebQSP | | CWQ | |
|---|---|---|---|---|
| | *Hit@1* | *F1* | *Hit@1* | *F1* |
| w/o $\mathcal{Z}_s$ | 93.9 | **64.8** | 74.0 | 45.3 |
| w/o $\mathcal{Z}_f$ | 86.2 | 56.8 | 79.5 | **46.9** |
| w/o $\mathcal{Z}_s, \mathcal{Z}_f$ | 57.6 | 60.0 | 59.7 | 38.0 |
| Ours | **95.4** | 63.3 | **80.8** | 45.0 |

outputs, disrupting normal language logic and damaging the model's original capabilities, also causing LLMs outputs to deviate from question logic. At $\omega = 2.0$, the best results are achieved across all metrics on all datasets, so we select $\omega = 2.0$ as the default strength value.

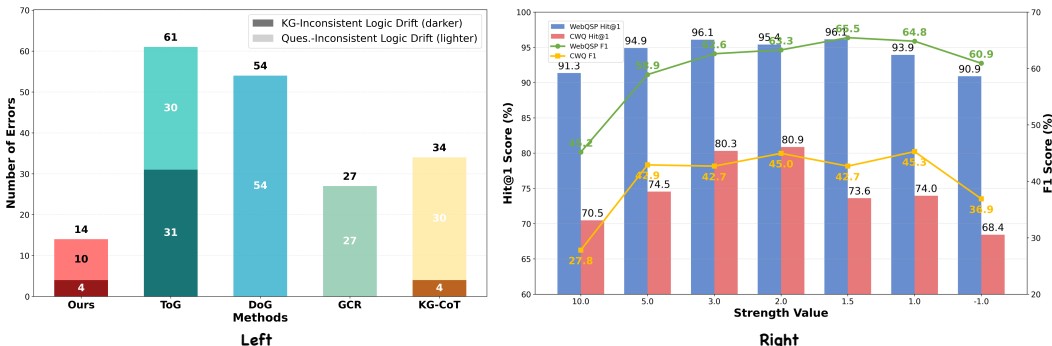

Figure 4: **Left:** Error analysis of Logits-to-Logic and advanced methods ToG, DoG, KG-CoT, and GCR. Lighter colors indicate Question-Inconsistent Logic Drift, while darker colors indicate KG-Inconsistent Logic Drift. **Right:** Impact of strength value in the logits strengthening module on reasoning performance.

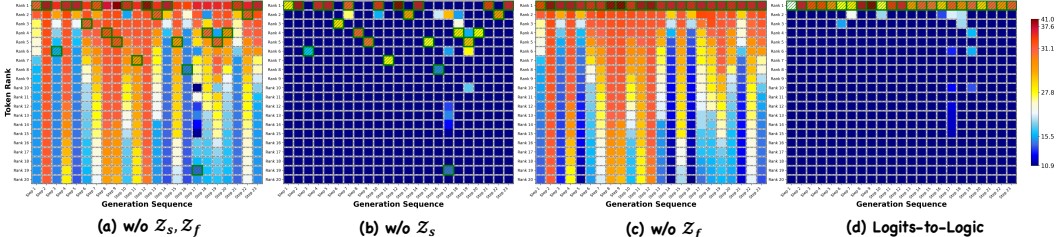

Figure 5: Visualization of LLMs output logits distribution and question, KG logical distributions. X-axis shows reasoning steps, Y-axis shows token logits rankings. Red colors indicate higher logits values. Green-bordered textured boxes are desired correct tokens (logically consistent with question and KG). We want green boxes to rank higher with redder colors.

We visualize LLMs' output logits distribution to intuitively explore whether the model performs logic-consistent reasoning. As shown in Fig. 5, colors represent logits values, with redder colors showing higher values and greater sampling probability. Green boxes represent correct tokens consistent with question logic (e.g, tokens in answer paths), while gray boxes represent incorrect tokens inconsistent with KG logic (e.g, tokens not in KG). Therefore, we expect green boxes to rank higher with redder colors, while gray boxes rank lower with bluer colors. Results show that (d) Logits-to-Logic achieves logic-consistent reasoning. Removing (c) logits filtering reveals high logits values for KG-inconsistent tokens (e.g, red areas in multiple gray boxes). Removing (b) logits strengthening shows some correct tokens ranking lower (e.g, decreased green box rankings).

Table 3: Transfer experiment of Logits-to-Logic. We transfer our methods to differecnt KGs and tasks. *, †, ‡ indicates w/ ChatGPT, GPT4, LLaMA3.1-8b, respectively.

| Method | Muti-hop | Slot Filling | |
|---|---|---|---|
| | QALD10-en | T-REx | Zero-shot RE |
| *LLMs Reasoning* | | | |
| IO prompt* | 42.0 | 33.6 | 27.7 |
| CoT* | 42.9 | 32.0 | 28.8 |
| SC* | 45.3 | 41.8 | 45.4 |
| *Agentic Reasoning* | | | |
| ToG-R* | 48.6 | 75.3 | 86.5 |
| ToG* | 50.2 | 76.8 | 88.0 |
| ToG-R† | 54.7 | 75.5 | 86.9 |
| ToG† | 53.8 | 77.1 | 88.3 |
| **Ours‡** | **59.2** | **87.5** | **91.3** |

Removing both (a) shows obvious logic drift with logits output inconsistent with question and KG logical distributions. This experiment intuitively demonstrates the important role of our core modules in maintaining LLMs' logic-consistent reasoning.

| Question | Which currency would be most useful where the newspaper *Akher Saa* circulated in 2013? |
|---|---|
| Reasoning Paths in KG | Akher Saa -> book.newspaper.circulation_areas -> Egypt -> location.statistical_region.gni_in_ppp_dollars -> g.124617kr
Akher Saa -> book.newspaper.circulation_areas -> Egypt -> location.country.currency_used -> Egyptian pound
Akher Saa -> book.periodical.language -> Arabic Language -> language.human_language.main_country -> Saudi Arabia |
| Answer Path | Akher Saa -> book.newspaper.circulation_areas -> Egypt -> location.country.currency_used -> Egyptian pound |
| LLMs Reasoning | Based on the question and the provided information Akher Saa -> book.periodical.language -> Arabic Language -> language.human_language.main_country -> Saudi Arabia, Akher Saa uses Arabic. Based on Arabic-speaking regions, the currencies that might be used to purchase this newspaper could be Sudanese Pound, Egyptian Pound, or South Sudanese Pound. |
| Agentic Reasoning | Based on the given information, I need to find details about Akher Saa's location. According to the KG showing Akher Saa -> book.periodical.language -> Arabic Language, it belongs to an Arabic-speaking region, therefore the answer is Arab -> people.country.currency_used -> South Sudanese Pound. |
| Logits-to-Logic w/o $\mathcal{Z}_s$ | Akher Saa -> book.newspaper.circulation_areas -> Egypt -> location.statistical_region.gni_in_ppp_dollars -> g.124617kr
Akher Saa -> book.newspaper.circulation_areas -> Egypt -> location.country.currency_used -> Egyptian pound |
| Logits-to-Logic w/o $\mathcal{Z}_f$ | Akher Saa -> book.newspaper.headquarters -> Arab -> people.country.currency_used -> South Sudanese Pound
Akher Saa -> book.newspaper.circulation_areas -> Egypt -> location.country.currency_used -> Egyptian pound |
| Logits-to-Logic | Akher Saa -> book.newspaper.circulation_areas -> Egypt -> location.country.currency_used -> Egyptian pound |

Figure 6: Case study of Logits-to-Logic.

Besides logits distribution visualization, we conduct detailed case analysis to illustrate logic drift and logic-consistent reasoning. As shown in Fig. 6, given the question *Which currency would be most useful where the newspaper Akher Saa circulated in 2013?*, the high-scoring answer path in NFA is *Akher Saa → book.newspaper.circulation_areas → Egypt → location.country.currency_used → Egyptian pound*. Both LLMs Reasoning and Agentic Reasoning consider the question-inconsistent path *Akher Saa → book.periodical.language → Arabic Language → language.human_language.main_country → Saudi Arabia* and infer the incorrect answer *Sudanese Pound* that does not exist in the KG. Therefore, both exhibit Logic Drift problems. Our method uses logits strengthening to enhance the path containing *Egyptian pound* and logits filtering to constrain non-KG paths, thereby aligning LLMs outputs with question and KG logic to achieve logic-consistent precise reasoning.

## 4.4 TRANSFER EXPERIMENTS (RQ3)

We conduct transfer experiment to explore the flexibility and robustness of the Logits-to-Logic framework. Without any modifications, our method easily transfers to unseen and larger-scale Wikidata-based datasets—QALD10-en, T-REx, and Zero-shot RE. Meanwhile, our method easily adapts to different structured knowledge reasoning tasks, including multi-hop QA, single-hop QA, and slot filling. Tab. 3 shows that our method maintains advanced performance across different tasks and KGs while preserving high flexibility, demonstrating the robustness of Logits-to-Logic.

## 4.5 DIFFERENT BACKBONE EXPERIMENTS (RQ4)

Table 4: Different backbones of Logits-to-Logic

| Backbone | WebQSP | | CWQ | |
|---|---|---|---|---|
| | Hit@1 | F1 | Hit@1 | F1 |
| *Microsoft series* | | | | |
| Phi-3-mini-4k | 83.1 | 65.7 | 69.5 | 50.2 |
| *Mistral AI series* | | | | |
| Mistral-7B | 95.2 | 61.8 | 77.2 | 48.3 |
| *Internlm series* | | | | |
| Internlm2_5-7b | 95.0 | 65.4 | 75.8 | 46.5 |
| Internlm2-7b | 95.6 | 65.3 | 75.6 | 51.7 |
| *Qwen series* | | | | |
| Qwen-2-0.5b | 81.6 | 60.4 | 75.2 | 35.9 |
| Qwen2-1.5b | 92.8 | 60.5 | 73.1 | 41.0 |
| Qwen-2-7b | 95.9 | 60.6 | 75.8 | 43.0 |
| Qwen2.5-14B | **96.6** | 71.5 | 77.0 | 54.8 |
| *Meta-llama series* | | | | |
| LLaMA-2-7b | 94.0 | 58.2 | 82.8 | 41.6 |
| LLaMA-3.1-8b | 95.4 | 63.3 | 80.9 | 45.0 |
| Llama-2-13b | 96.0 | **75.2** | **83.1** | **64.1** |

We explore the impact of different LLM backbones on reasoning performance.

We mainly select five mainstream open-source LLM series: (1) Meta-LLaMA (Touvron et al., 2023), (2) Qwen (Bai et al., 2023), (3) Microsoft (Abdin et al., 2024), (4) Mistral AI (Jiang et al., 2023a) and (5) Internlm series (Cai et al., 2024). We compare open-source LLMs from 0.5b to 13b parameters. As shown in Tab. 4, LLaMA-2-13b achieves the highest F1 score, while Qwen and LLaMA 13b models achieve the highest Hit@1 scores on two datasets respectively, with small differences from the former. Therefore, considering overall performance and efficiency, we select LLaMA-3.1-8b as the base model. Meanwhile, smaller 0.5b and 1.5b models also achieve excellent performance under the Logits-to-Logic framework.

### 4.6 Error Analysis (RQ5)

We conduct detailed error analysis of our method, as shown in Fig. 4 **Left**. We mainly analyze the two error types mentioned in the Sec. 1: (1) Question-Inconsistent Logic Drift; (2) KG-Inconsistent Logic Drift. We compare with advanced methods including ToG, DoG, KG-CoT, and GCR. Results intuitively show that Logits-to-Logic has the lowest error number of 14, effectively reducing logic-inconsistent reasoning errors. Meanwhile, Question-Inconsistent Logic Drift count is only 10 in Logits-to-Logic, while other methods all exceed 27. Results demonstrate that our method effectively enhances LLMs' logic-consistency in structured knowledge reasoning.

## 5 Conclusion and Future Work

In this work, we propose a flexible structured knowledge reasoning approach *Logits-to-Logic*. We highlight that the key challenge of LLMs in structured KG reasoning lies in the inconsistency between their outputs and the logical distributions of KG and question. Unlike previous work focusing solely on input, we propose addressing Logic Drift from the output perspective. We unify the LLM's autoregressive generation and the KG's structure within a state-transition NFA, and introduce logits strengthening and logits filtering to mitigate Logic Drift, thereby achieving precise logical reasoning. Extensive experiments demonstrate that *Logits-to-Logic* achieves state-of-the-art performance in structured knowledge reasoning while maintaining flexibility and robustness for transfer across different KGs and tasks.

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

## A    RELATED WORK

Recent advances in knowledge graph question answering have predominantly adopted agentic paradigms to struggle to address the logic drift issues of large language models in structured knowledge reasoning, with approaches broadly categorized into path-based reasoning and KG-constrained generation methods.

**Path-based Reasoning Approaches.**    Path-based methods focus on decomposing complex queries into structured reasoning paths over knowledge graphs. KG-CoT (Zhao et al., 2024) uses collaborative frameworks where small models generate candidate KG paths as constraints while large models make final decisions, though this approach suffers from quality dependencies and potential logic drift in path selection. DoG (Ma et al., 2025) employs a three-role multi-agent debate framework with iterative decomposition for enhanced logical consistency, though it lacks explicit KG path constraints and remains sensitive to role design. RoG (Luo et al., 2024) advances this direction through agent-based planning that generates candidate paths with structural constraints, improving interpretability but requiring costly model fine-tuning with poor cross-KG transferability. ToG (Sun et al., 2023) further refines path-based reasoning by decomposing multi-hop queries into single-hop operations with explicit path construction, yet remains susceptible to error accumulation and logic drift from sequential path selection. Similarly, DARA (Fang et al., 2024) converts multi-hop reasoning into structured SPARQL query processes through multi-agent decomposition, but lacks guarantees for logical compliance between generated queries and KG structure.

**KG-Constrained Generation Methods.**    KG-constrained approaches emphasize incorporating structural knowledge constraints during the generation process. GCR (Luo et al., 2025) introduces dual-agent reasoning combining KG experts with LLMs, using KG Trie structures to constrain generation to verified paths, but KG experts may propose contextually inconsistent paths. PoG (Chen et al., 2024) and GoG (Xu et al., 2024) represent prompt-based constraint methods, with PoG embedding "retrieve-reason" workflows for gradual KG exploration and GoG combining structured retrieval with parametric knowledge through "Thought-Action-Observe" reasoning, though both lack hard constraints and verification mechanisms. KG-Agent (Jiang et al., 2025) systematizes constraint-based reasoning through specialized planner, toolbox, and executor roles, but struggles with generalization across diverse KG logical patterns.

**Convergence on Agentic Paradigms.**    Notably, both path-based and KG-constrained approaches converge on agentic paradigms, employing multi-agent frameworks, role-based decomposition, and iterative reasoning processes to bridge the gap between unstructured language model capabilities and structured knowledge graph reasoning requirements. Existing work mainly addresses logic drift by designing increasingly complex agent-based frameworks. However, in summary, most advanced methods embed complex, task-specific workflows in prompts, offering only input-level guidance that neither resolves logic drift fundamentally in the output nor provides robust constraint enforcement across diverse knowledge graph domains.

Unlike previous methods that design complex workflows or prompt engineering, we address Logic Drift from the output perspective, providing deeper insights into logic-consistent reasoning.

## B    DATASETS STATISTICS

As shown in Table 5, we evaluate our method's performance across multiple benchmark datasets and various tasks. Specifically, we comprehensively assess the method's reasoning capabilities across different tasks and examine its robustness on knowledge graphs of varying scales.

**Tasks.**    We design three different types of reasoning tasks: Multi-hop QA, Single-hop QA, and Slot Filling. Multi-hop datasets primarily test the method's multi-step reasoning capabilities, requiring the model to master complex logical chains and navigate through multiple knowledge graph relations to arrive at the correct answer. For example, a multi-hop question like *Where did the headliner of the Jay Z 2009 Concert Tour grow up?* requires the reasoning path: *Jay Z 2009 Concert Tour →  music.concert_tour.artist → Jay-Z → people.person.place_of_birth → Brooklyn*. Among these, we focus particularly on CWQ (ComplexWebQuestions) and WebQSP as our primary evaluation datasets,

| Dataset | # Train | # Test | Answer Format | Task | Background KG |
|---------|---------|--------|---------------|------|---------------|
| ComplexWebQuestions | 27,734 | 3,531 | Entity | Muti-hop QA | Freebase |
| WebQSP | 3,098 | 1,639 | Entity/Number | Muti-hop QA | Freebase |
| GrailQA* | 44,337 | 1,000 | Entity/Number | Muti-hop QA | Freebase |
| QALD-10 | - | 333 | Entity/Number | Muti-hop QA | Wikidata |
| Simple Quesiton* | 14,894 | 1,000 | Entity/Number | Single hop QA | Freebase |
| T-REx | 2,284,168 | 5,000 | Entity | Slot Filling | Wikidata |
| Zero-Shot RE | 147,909 | 3,724 | Entity | Slot Filling | Wikidata |

Table 5: Overview of dataset statistics used in this study. * means we randomly chose 1,000 samples from the GrailQA and Simple Questions test set to create the testing set because of the abundant test samples.

as they impose higher demands on multi-hop reasoning capabilities and better reflect the model's logical consistency in structured knowledge reasoning. Single-hop QA datasets evaluate the model's ability to directly retrieve information through simple, one-step reasoning processes, such as *Who played in the Forbidden Zone and is the voice of Jack Skellington?* which follows the path *Forbidden Zone → film.film.music → Danny Elfman*. Slot Filling tasks assess the model's capacity to extract and fill missing information in structured knowledge representations, for instance, given *Egelsee [SEP] country*, the model should identify the answer as *Switzerland, Austria, Germany*.

**Knowledge Graphs.** Following previous work, we use Freebase and the larger-scale Wikidata to examine our method's robustness. Freebase serves as a substantial knowledge graph with 88 million entities, 20,000 relations, and 126 million triples, providing a comprehensive testbed for evaluating reasoning performance on well-structured, curated knowledge. Wikidata, as one of the largest publicly available knowledge graphs containing over 100 million entities and billions of statements, allows us to assess our method's scalability and effectiveness when dealing with massive, real-world knowledge repositories. The choice of these two knowledge graphs enables us to evaluate our approach across different scales. This dual evaluation setup ensures that our method demonstrates consistent performance across varying knowledge graph complexities and scales.

## C   MAIN RESULTS OF F1

To comprehensively demonstrate the performance of our method, we supplement with F1 score comparison experiments in Tab. 6. Since some agentic reasoning methods did not report F1 metrics in their papers, we present comparisons with methods that reported F1 metrics including GNN-RAG (Mavromatis & Karypis, 2025), GSR (Huang et al., 2024), Interactive-KBQA (Xiong et al., 2024), SubgraphRAG (Li et al., 2025) (data sourced from their respective papers). Combined with the experimental results in Tab. 1, our method demonstrates comprehensive advantages in both Hit@1 and F1 scores.

## D   IMPLEMENTATION DETAILS

**Data Preparation.** We preprocess all valid paths in the knowledge graph by compiling them into a Non-deterministic Finite Automaton (NFA) in an offline manner. For each given dataset, the original questions and their corresponding topic entities are provided. For every sample in the dataset, we extract a 2-hop subgraph related to the question from the corresponding background knowledge graph (either Freebase or Wikidata). Specifically, starting from the topic entity, we employ Breadth-First Search (BFS) to explore all 2-hop paths in the vicinity of the topic entity. These discovered paths serve as the valid acceptable paths $S$ within our NFA. We then convert these paths into textual representations by connecting entities and relations using the $\rightarrow$ delimiter. Subsequently, we utilize the LLM's tokenizer to parse each path into token sequences, treating all possible token subsequences as valid states $S_{0:end}$. For instance, given a path *Help Me Make It Thru the Night → music.composition.composer → Joe Walsh* in set $S$, both *Help Me Make It Thru the Night → music.composition.composer → Joe* and *Help Me Make It Thru the Night → music.composition.composer* are considered valid states in our automaton. For all acceptable paths in $S$, we employ a lightweight sentence-transformer model (sentence-transformer-all-MiniLM-L6-v2

Table 6: F1 Performance comparison of Logits-to-Logic and various baselines on CWQ and WebQSP datasets, with the **best results in bold**.

| Method | WebQSP | | CWQ | |
|---|---|---|---|---|
| | F1 | Hit@1 | F1 | Hit@1 |
| *Closed-source business model* | | | | |
| ChatGPT | 43.5 | 59.3 | 30.2 | 34.7 |
| ChatGPT+Few-shot | 38.1 | 68.5 | 28.0 | 38.5 |
| ChatGPT+CoT | 38.5 | 73.5 | 31.0 | 47.5 |
| *Hybrid System (Large-Small Model Collaboration)* | | | | |
| GCR w/ LLaMA3-8b+ChatGPT | 73.2 | 92.6 | 60.9 | 72.7 |
| GCR w/ LLaMA3-8b+GPT4o-mini | 74.1 | 92.2 | 61.7 | 75.8 |
| GNN-RAG w/ LLaMA2-7b+GNN | 71.3 | 80.6 | 59.4 | 61.7 |
| GSR w/ T5-3b + LLaMA2-7b | 58.9 | - | 27.6 | - |
| *7-8b model* | | | | |
| Qwen2-7b | 35.5 | 50.8 | 21.6 | 25.3 |
| LLaMA2-7b | 36.5 | 56.4 | 21.4 | 28.4 |
| LLaMA3.1-8b | 34.8 | 55.5 | 22.4 | 28.1 |
| Interactive-KBQA w/ Mistral-7B | 43.5 | 45.0 | 39.9 | 44.0 |
| KD-CoT w/ LLaMA2-7b | 52.5 | 68.6 | - | 55.7 |
| RoG w/ LLaMA2-Chat-7B | 70.8 | 85.7 | 56.2 | 62.6 |
| SubgraphRAG w/ Llama3.1-8B | 67.9 | 81.2 | 43.0 | 47.5 |
| Logits-to-Logic w/ LLaMA3-8b | 63.3 | 95.4 | 45.0 | 80.9 |
| *13b model* | | | | |
| Interactive-KBQA w/ LLaMA2-13b | 54.8 | 56.2 | 42.5 | 45.6 |
| RoG w/ LLaMA3-13b | 73.6 | 89.1 | 63.2 | 68.3 |
| Logits-to-Logic w/ LLaMA3-13b | **75.2** | **96.0** | **64.1** | **83.1** |

with only 22M parameters) to generate embeddings and compute semantic similarity scores between each path and the question embedding. We select the path with the highest similarity score as the top-1 candidate, which will be masked in the MASK-Prompt during the logits strengthening process.

**Output Format.** We observe that smaller-scale LLMs cannot reliably adhere to specific output formats through zero-shot or few-shot prompting approaches, which complicates the extraction of reasoning paths and answers from LLM outputs during evaluation. To ensure that LLM outputs conform to our textual format requirements (i.e., using $\to$ to connect entities and relations), we perform supervised fine-tuning using 10% of randomly sampled data from the WebQSP and CWQ training sets. This fine-tuning approach serves solely to teach the LLMs the correct output format without exposing them to unseen knowledge from the test set, as the data is strictly segregated to prevent information leakage. The fine-tuning process focuses exclusively on format compliance rather than knowledge acquisition.

**Logic-Consistent Reasoning.** We implement our method using PyTorch on Ubuntu 20.04.1 LTS servers. During the inference phase, our method requires only 16GB memory per single card for LLaMA3-8B inference (batch size=1, fp16, context length 2048) and can run efficiently on a single GPU with memory capacity greater than 32GB (Details in Appendix I). While our experimental setup uses two A800 80G GPUs for higher throughput, no model parallelism or cross-GPU communication is required.

# E    DETAILS OF BASELINES

We compare two mainstream categories of methods: (1) LLMs Reasoning methods use prompt engineering with LLMs for structured knowledge reasoning; (2) Agentic Reasoning methods treat KGs as dynamic environments and design intricate prompts and workflows to guide multi-agent collaborative reasoning. The following provides detailed introductions to the baselines of these two categories of methods.

**LLMs Reasoning**   methods use prompt engineering with LLMs for structured knowledge reasoning.

- IO (Brown et al., 2020b) prompt is the most basic approach that directly feeds questions to ChatGPT without reasoning guidance, relying solely on pre-trained knowledge. Its main limitation is the lack of explicit reasoning direction, making it difficult to handle complex multi-step knowledge graph queries and often producing logically inconsistent responses.

- CoT (Wei et al., 2022) methodology enhances reasoning by guiding large language models to generate step-by-step processes, decomposing complex questions into intermediate steps and building reasoning chains. The approach incorporates example reasoning processes within prompts, demonstrating logical progression from question to answer. However, CoT's improvement in knowledge graph reasoning remains limited due to lacking explicit constraints on structured knowledge, causing reasoning chains to potentially deviate from actual knowledge graph.

- SC (Self-Consistence) (Wang et al., 2023b) method refines CoT by generating multiple different reasoning paths and selecting the most consistent answer. The approach instructs ChatGPT to create multiple distinct reasoning chains for the same question, then determines the final answer through voting mechanisms or consistency checking. The core principle uses diversity sampling to reduce errors in single reasoning attempts.

**Agentic Reasoning**   methods treat KGs as dynamic environments and design intricate prompts and workflows to guide multi-agent collaborative reasoning.

- StructGPT (Jiang et al., 2023b) enhances LLMs' reasoning over structured data using an Iterative Reading-then-Reasoning (IRR) approach, which includes specialized interfaces for efficient data access, a novel invoking-linearization-generation procedure, and iterative reasoning to effectively utilize structured data in answering complex questions.

- KD-CoT (Wang et al., 2023a) extends standard chain-of-thought prompting with an explicit retrieval loop. At each iteration the model first generates a "Thought" that decomposes the original query into a focused sub-question; it then performs an "Action" by querying the external knowledge base to fetch facts relevant to that sub-question. The newly retrieved evidence is appended to the context, allowing the model to refine its reasoning and repeat the cycle until a complete multi-hop answer is assembled.

- KG-CoT (Zhao et al., 2024) designs collaborative structured knowledge reasoning between large and small models, using small agents to perform reasoning on the KG to obtain candidate paths, then employing large LLMs to make decisions based on the candidate paths.

- RoG (Luo et al., 2024) performs reasoning path generation through agent planning and prediction. RoG requires fine-tuning the model to plan and generate candidate path sets, then selects reasoning paths from these candidates. However, it needs to acquire KG knowledge through fine-tuning, making it difficult to transfer across different KGs and tasks.

- DoG (Ma et al., 2025) designs three specialized agent roles (simplify, critic, linguist) to iteratively decompose complex questions and correct reasoning logic through single-step modifications. This multi-agent framework leverages the distinct capabilities of each role to address different aspects of the reasoning process, enabling structured knowledge reasoning through collaborative agent interactions and iterative refinement of the reasoning chain.

- ToG (Sun et al., 2023) designs step-by-step reasoning processes, enhancing LLMs' understanding of structured knowledge logic through single-step entity and relation exploration on KGs. This approach breaks down complex multi-hop queries into manageable single-hop operations, allowing the model to progressively build reasoning paths while maintaining awareness of the underlying graph structure throughout the reasoning process.

- GCR (Luo et al., 2025) adopts a dual-agent scheme that pairs a knowledge-graph specialist with a large prediction model. The specialist, constrained by a KG Trie, is only allowed to propose paths that actually exist in the graph, thereby preventing it from hallucinating nonexistent relations or entities. These verified paths are then passed to the larger LLM, which performs the final reasoning over the confirmed route and produces the answer.

- PoG (Chen et al., 2024) introduces an iterative "Retrieve–Reason" workflow that incrementally explores the knowledge graph; this workflow is embedded in the prompt to steer the LLM through the reasoning process.
- GoG (Xu et al., 2024) designs a refined agent reasoning workflow "Thought-Action-Observe", using the model's internal parametric knowledge to compensate for the deficiencies of incomplete KGs, enhancing interpretability and confidence.
- DARA (Fang et al., 2024) employs LLM agents to progressively decompose complex questions step-by-step, solving the entire problem through iterative generation of SPARQL queries for sub-questions. This approach systematically breaks down complex multi-hop reasoning tasks into manageable components, enabling structured query formulation.
- KG-Agent (Jiang et al., 2025) designs specialized planner, toolbox, and executor roles for automated reasoning in knowledge graphs, enabling systematic decomposition and execution of complex queries. However, their custom toolboxes cannot comprehensively cover all logical patterns present in diverse knowledge graphs, limiting the method's ability to handle the full spectrum of reasoning scenarios.
- Sym-Agent (Liu et al., 2025) designs a self-learning agent reasoning workflow to enhance structured knowledge reasoning, constructing tool-calling reasoning trajectories to help agents learn and reflect on their reasoning processes. This approach enables continuous improvement through iterative learning cycles, where agents analyze their previous reasoning steps and refine their strategies based on feedback and performance evaluation across different reasoning scenarios.

# F THEORETICAL DERIVATION OF LOGITS-TO-LOGIC OPTIMIZATION OBJECTIVE

Given question $q$ and knowledge graph $G$, our goal is to enable LLMs to perform logic-consistent reasoning, which helps derive answer paths $s_+^{e^{topic}}$:

$$\left\{ s_+^{e^{topic}} \right\} \propto \mathcal{D}_{q,G} \sim \underset{\mathcal{D}_\theta}{argmax} P_\theta \left( a|q,G \right)$$

In general, our reasoning objective during decoding is: using $\mathcal{Z}_s$ and $\mathcal{Z}_f$ to align LLMs' logits with the logical distributions of $q$ and $G$ ($s_+^{e^{topic}} \cup s_-^{e^{topic}}$), making LLMs output correct answer paths $s_+^{e^{topic}}$ while avoiding incorrect paths $s_-^{e^{topic}}$.

$$P_\theta \left( a \mid q, G \right) \propto P_{\theta,q,G} \left( a \mid q, G \right) = P_{\theta,q,G} \left( a \mid q, s_+^{e^{topic}}, s_-^{e^{topic}} \right)$$

The core of the knowledge graph $G$ lies in the "set of positive and negative samples of topic-related entities", i.e., $G = \left\{ s_+^{e^{topic}}, s_-^{e^{topic}} \right\}$ (Positive samples $s_+$ are valid information related to the question's topic entity $e^{topic}$, while negative samples $s_-$ are irrelevant interfering information).

The generation of answer $a$ relies solely on the positive sample set $s_+^{e^{topic}}$ and is independent of the negative samples $s_-^{e^{topic}}$. As irrelevant interference items, negative samples do not provide effective logical support for answer $a$ (e.g., response generation, decision-making). They can thus be excluded from the conditions, leading to:

$$P_{\theta,q,G}(a \mid q, \{s_+^{e^{topic}}\}, \{s_-^{e^{topic}}\}) \propto P_\theta(a \mid q, \{s_+^{e^{topic}}\})$$

The generation of positive samples $s_+^{e^{topic}}$ is driven by two independent logics:

**Question logic** : The semantic requirements of the question $q$ itself (e.g., "asking for Elon Reeve Musk's place of birth" requires associating positive samples related to "Elon Reeve Musk"), corresponding to the distribution $\mathcal{D}_q \sim P_{\theta,q}(s_+^{e^{topic}} \mid q, G)$;

**Knowledge graph logic** : The inherent associations of entities in the KG (e.g., the "place of birth" association between "Elon Reeve Musk" and "Pretoria" in the KG), corresponding to the distribution $\mathcal{D}_G \sim P_{\theta,G}(s_+^{e^{topic}} \mid q, G)$.

Since these two logics are independent (the question's semantic requirements have no direct dependence on the KG's inherent structure), the joint probability of positive samples is the product of the two according to the multiplication rule for independent events:

$$P_{\theta,q,G}(\{s_+^{e^{topic}}\} \mid q, G) = P_{\theta,q}(\{s_+^{e^{topic}}\} \mid q, G) \cdot P_{\theta,G}(\{s_+^{e^{topic}}\} \mid q, G)$$

Combining the "dependence of answer $a$ on positive samples" and the "probability decomposition of positive samples", and applying the chain rule of probability $P(A \mid B, C) \propto P(A \mid C) \cdot P(C \mid B)$ $(where \ A = a, \ B = q, G, \ C = s_+)$, we finally obtain:

$$\mathcal{D}_{q,G} \sim P_{\theta,q,G}(a \mid q, G) \propto P_\theta(a \mid q, \{s_+^{e^{topic}}\}) \cdot \mathcal{D}_q \cdot \mathcal{D}_G$$

## G  PROMPT DETAILS

Figure 7: Prompt template of Logits-to-Logic.

As shown in Fig. 7, our prompt consists of two components: `INSTRUCTION` and `INFORMATION`. The `INSTRUCTION` part describes the role and responsibilities of LLMs and provides task instructions to users. We need LLMs to generate precise reasoning paths based on structured KG information. The `INFORMATION` part includes: (1) the question; (2) the topic entity corresponding to the question; (3) paths in the KG. Specifically, as mentioned in Sec. 3.2.2, we use a score model to evaluate all paths (as shown by the red scores on the right side of each path, which are for illustration purposes only and do not appear in the actual prompt), and mask the high-scoring paths (highlighted with green background: *Help Me Make It Thru the Night → music.composition.composer → Joe Walsh → music.guitarist.guitars_played → Fender Stratocaster*) to obtain the MASK-Prompt.

## H    ALGORITHM OF LOGITS-TO-LOGIC

We detail the algorithm of Logits-to-Logic in Algorithm 1.

---

**Algorithm 1:** Logits-to-Logic Reasoning

---

**Input**   : Question $q$, knowledge graph $G$, LLMs $M_\theta$, vocabulary $\Sigma$, score model $M_\Phi$, strength value $\omega$,
   beam size $b_{nums}$

**Output** : Prediction path $P$

$P \leftarrow []$;

\# **Step 1: Logic Compiling**

$e^{topic} \leftarrow$ Extract topic entity from $q$;

$S \leftarrow \text{BFS}\,(e^{topic}, 2-hop) \leftarrow$ Extract all paths from $G$ starting from $e^{topic}$;

$S_{0:end} \leftarrow$ Decompose all possible states of $S$;

$\delta \leftarrow$ Set the state transition function such that: $\delta(t) = t \times S_{i:end} \to S_{i+1:end}, t \in \Sigma$;

$NFA = \left(S_{0:end}, \Sigma, \delta, e^{topic}, S\right) \leftarrow$ Build NFA;

$S \leftarrow \text{score}(M_\Phi, q, S)$ Using the score model $M_\Phi$ we compute the semantic similarity between $q$ and $S$
   to obtain $S$ with score;

$NFA_\Phi = (S_{0:end}, \Sigma, \delta, e^{topic}, S) \leftarrow$ Get $NFA_\Phi$;

**foreach** $b \in b_{nums}$ **do**
    $GenSeq \leftarrow []$;
    \# $i+1-th$ token generation
    **while** $GenSeq.end()\,! = eos.token$ **do**
        \# **Step 2: Logits Strengthening**
        \# Filter high-scoring and low-scoring paths in $S$ within the $NFA$
        $s_+, s_- \leftarrow \text{Filter}\,(S),\ S\ in\ NFA_\Phi$;
        Prompt $\leftarrow \text{Texualize}\,(\text{INSTRUCTION}, s_+, s_-)$;
        MASK-Prompt $\leftarrow \text{Texualize}\,(\text{INSTRUCTION}, \text{MASK}, s_-)$;
        logits z distribution $P_{\theta, \mathcal{Z}_s}(s_+^{e^{topic}}|q, G) \sim \mathcal{D}_q \leftarrow \{M_\theta(\text{Prompt}) - M_\theta(\text{MASK-Prompt})\} * \omega$;
        \# **Step3: Logits Filtering**
        logits z distribution $P_{\theta, \mathcal{Z}_s, \mathcal{Z}_f}(s_+^{e^{topic}}|q, G) \sim \mathcal{D}_{q,G} \leftarrow \mathcal{D}_q * \delta(t_{i+1}, S_{i:end}),\ S_{i:end}\ in\ NFA_\Phi$;
        \# Sample to obtain tokens
        $t_{i+1} \leftarrow \text{Sample}(\mathcal{D}_{q,G})$;
        GenSeq.append($t_{i+1}$);
    $P.append(GenSeq)$;

**return** $P$;

---

## I    COMPUTATION COST OF LOGITS-TO-LOGIC

As shown in Tab. 7, we conducted experiments on the computational overhead of the core modules $\mathcal{Z}_s$ and $\mathcal{Z}_f$ in Logits-to-Logic. The experiments show that $\mathcal{Z}_f$ improves decoding speed by restricting the generation of illegal tokens and reducing the exploration space during LLM decoding, while $\mathcal{Z}_s$ requires an additional forward computation to obtain the logits distribution of the mask prompt, thus

Table 7: Computational overhead of the core modules in Logits-to-Logic.

| Method | GPU Usage (GB) | Running Time (h) | |
|---|---|---|---|
| | | CWQ | WebQSP |
| LLaMA3-8b | 29.01 | 15.28 | 7.25 |
| LLaMA3-8b w/ $\mathcal{Z}_f$ | 29.01 | 8.11 | 6.41 |
| LLaMA3-8b w/ $\mathcal{Z}_s$ | 29.09 | 26.33 | 11.06 |
| LLaMA3-8b w/ Logits-to-Logic | 32.89 | 18.63 | 8.48 |

Table 8: Comparison of computational cost with other methods, including the number of LLM API calls, expenses, and the number of tokens consumed per question. Note: Logits Strengthening involves two forward passes per decoding step but is implemented as a single generate call using internal callbacks, appearing as one LLM call per question.

| Model | CWQ | | | WebQSP | | |
|---|---|---|---|---|---|---|
| | # LLM Call | Total Token | Total Cost ($) | # LLM Call | Total Token | Total Cost ($) |
| *LLMs Reasoning* | | | | | | |
| CoT | 1 | 409.7 | 0.00008 | 1 | 397.6 | 0.00008 |
| *Agentic Reasoning* | | | | | | |
| ToG | 9.2 | 11468.5 | 0.0023 | 8.8 | 10189.4 | 0.0021 |
| DoG | 5.7 | 37919.7 | 0.006 | 2.7 | 6114.5 | 0.001 |
| GCR | 2 | 125.3 | - | 2 | 231 | - |
| KG-CoT | - | - | - | 1 | - | 0.02 |
| PoG | 13.3 | 8,156.2 | 0.0016 | 9 | 5,517.7 | 0.0011 |
| Logits-to-Logic | 1 | 1270.8 | - | 1 | 716.9 | - |

introducing a small computational overhead. On LLaMA3-8b, our method only requires an additional (∼3GB) memory usage, and adds 3 hours and 1 hour of runtime on CWQ and WebQSP respectively.

As shown in Tab. 8, we conduct a comprehensive comparison of computational overhead between Logits-to-Logic and state-of-the-art methods. The results demonstrate that our approach exhibits substantial advantages in terms of LLM API call frequency and computational expenses compared to existing methods.

When compared with ToG, DoG, and PoG, our method demonstrates comprehensive superiority in total token consumption. Specifically, on the CWQ dataset, Logits-to-Logic achieves remarkable reductions of 89%, 96%, and 84% in token consumption compared to ToG, DoG, and PoG respectively. These significant reductions in token usage translate directly to substantial cost savings and improved computational efficiency, making our approach more practical for large-scale deployment and real-world applications. Furthermore, our method shows exceptional efficiency on the WebQSP dataset. In comparison with ToG, DoG, KG-CoT, and PoG, our approach requires no additional computational overhead for reasoning tasks on WebQSP, demonstrating its ability to perform complex reasoning without incurring extra costs. This zero additional overhead characteristic represents a significant advancement in computational efficiency for knowledge graph reasoning tasks. When benchmarked against GCR, our method maintains optimal efficiency by requiring only one LLM API call per question. This minimal API usage not only reduces computational costs but also decreases latency and improves response times, making the system more responsive and scalable. The single-call requirement represents a substantial improvement over methods that necessitate multiple iterative calls to achieve comparable reasoning performance.

These comprehensive results collectively demonstrate that Logits-to-Logic possesses significant computational overhead advantages across multiple dimensions, establishing it as a more efficient and cost-effective solution for structured knowledge reasoning tasks.

## J  COMPUTATIONAL COST OF CONSTRUCTING NFA

In this section, we analyze the computational complexity of constructing NFAs in Stage 1 Logic Compiling (Sec. 3.2.1) through both theoretical and experimental analysis.

**Theoretical Analysis**  : We employ BFS to explore KG. Given that each question's topic entity $e^{topic}$ belongs to $E$, the average number of paths is $R^D$, where $R$ represents the average number of relations per entity $E$, and $D$ denotes the exploration depth (e.g., $D = 2$ for exploring 2-hop paths). With an average of $N_T$ tokens per path, the computational complexity for constructing NFAs per question is $N_T * R^D$. During LLM generation with beam size $N_B$, the LLM computational complexity becomes $N_B * N_T * R^D$.

**Experimental Analysis** : We collected and report path statistics for the CWQ and WebQSP datasets using Freebase as the background KG, as shown in Tab. 9.

Table 9: The path statistics for the CWQ and WebQSP datasets using Freebase as the background KG.

| Value | CWQ | | | WebQSP | | |
|---|---|---|---|---|---|---|
| | Min | Max | Avg | Min | Max | Avg |
| $R$ (relation num per entity) | 0 | 259 | 43.6 | 2 | 258 | 76.9 |
| $D$-hop size ($D = 2$) | 9 | 11011 | 2025.6 | 10 | 11713 | 2024.6 |
| $N_T$ ($D = 2$ per path token length) | 13 | 524 | 29.7 | 13 | 524 | 20.9 |

CWQ and WebQSP require exploring approximately 2000 2-hop paths per question on average, with average token lengths of 29.7 and 20.9 per path, respectively.

We report the detailed machine configuration used in our experiments in Tab. 10.

Table 10: Configuration of our machine.

| Machine Info. | Value |
|---|---|
| OS | Ubuntu 20.04.1 LTS |
| CPU | Intel Xeon Gold 6326, 64 cores, 40 threads, 2.90GHz |
| Memory | 629GB DDR4, 2666MHz |

We measured the computational overhead of NFA construction for 2-hop paths in Tab. 11, 12.

Table 11: Computational overhead of NFA construction for 2-hop paths in CWQ dataset.

| # Thread | CWQ(# samples=27639) | | |
|---|---|---|---|
| | Avg. Running Time per sample (s) | Peak Memory Usage (GB) | Parallel Efficiency (%) |
| 1 | 0.046 | 103.7 | 127 |
| 2 | 0.042 | 103.7 | 152 |
| 4 | 0.039 | 103.7 | 159 |
| 8 | 0.038 | 103.7 | 166 |
| 16 | 0.038 | 103.7 | 166 |
| 32 | 0.036 | 103.8 | 168 |

Table 12: Computational overhead of NFA construction for 2-hop paths in WebQSP dataset.

| # Thread | WebQSP (# samples=2826) | | |
|---|---|---|---|
| | Avg. Running Time per sample (s) | Peak Memory Usage (GB) | Parallel Efficiency (%) |
| 1 | 0.052 | 12.0 | 133 |
| 2 | 0.049 | 12.0 | 165 |
| 4 | 0.045 | 12.0 | 175 |
| 8 | 0.043 | 12.0 | 178 |
| 16 | 0.042 | 12.0 | 189 |
| 32 | 0.039 | 12.0 | 192 |

The experiments demonstrate the feasibility of constructing NFAs on large-scale KGs without requiring high time and computational overhead.

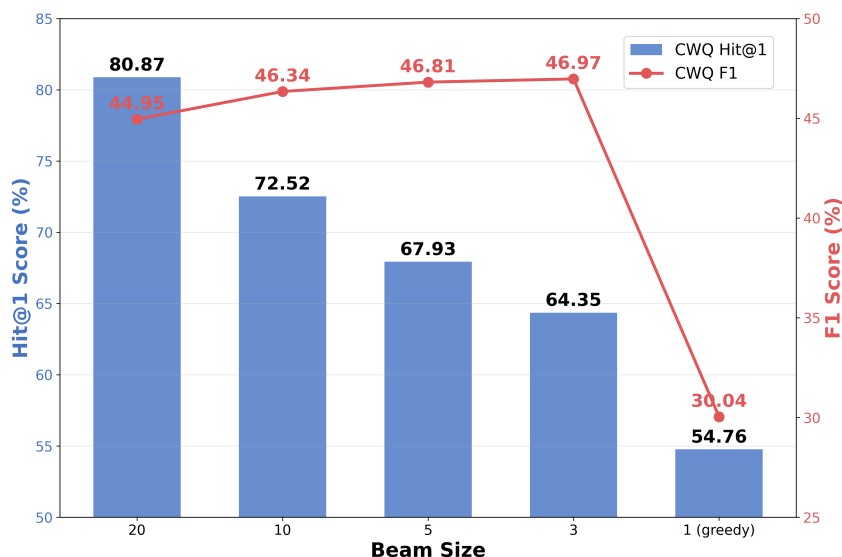

Figure 8: Impart of beam size.

## K    IMPACT OF BEAM SIZE

We conduct ablation experiments on different beam size values using the CWQ dataset, with results presented in Fig. 8. The experimental findings reveal several important insights regarding the relationship between beam size and model performance across different evaluation metrics.

When the beam size is set to 1, this configuration is equivalent to having the LLMs execute a greedy search strategy, where only the single most probable path is considered. As the beam size value increases, we observe a gradual improvement in the Hit@1 score, with optimal performance achieved when the beam size reaches 20. This improvement can be attributed to the expanded search space that allows the model to explore multiple promising reasoning paths simultaneously, thereby increasing the likelihood of identifying the correct reasoning paths. However, our analysis reveals a trade-off between different performance metrics as the beam size increases. While the Hit@1 score improves with larger beam sizes, the F1 score exhibits a declining trend. This phenomenon occurs because the expansion of the search space increases the probability that correct reasoning paths will be discovered, leading to improved answer recall rates. Nevertheless, this broader search inevitably introduces some incorrect reasoning paths into the candidate set, which consequently reduces the precision of predictions and results in an overall decrease in F1 scores. The underlying mechanism behind this trade-off lies in the fundamental tension between exploration and precision. A larger beam size enables more comprehensive exploration of the reasoning space, capturing more potential correct answers (higher recall), but simultaneously admits more false positives (lower precision). This behavior is consistent with typical beam search characteristics in sequence generation tasks, where broader search spaces often lead to improved coverage at the expense of precision.

Given that the F1 scores for beam sizes of 10 and 20 are approximately equivalent, we prioritize maintaining overall prediction quality and select beam size = 20 as the optimal configuration. This choice represents a balanced approach that maximizes the Hit@1 performance while maintaining acceptable F1 scores, ensuring that our method achieves both high accuracy in top-1 predictions and reasonable overall quality in the complete set of generated reasoning paths.

## L    DISCUSSION ON DIFFERENCES BETWEEN LOGIC DRIFT AND HALLUCINATION

Logic drift is a specific type of hallucination phenomenon in Knowledge Graph Question Answering (KGQA). Hallucination is more broadly defined as the output of LLMs that is inconsistent with facts,

logic, or given context, including the generation of false numbers, dates, names, or various incorrect outputs such as failing to follow instructions.

Although structured knowledge such as knowledge graphs has been introduced to reduce the hallucination problem of LLMs, in practical applications, the reasoning output of LLMs still exhibits inconsistency with the question intent and the logic of credible knowledge in the knowledge graph. The specific manifestations are: outputting reasoning paths irrelevant to the question intent, or outputting paths in the knowledge graph that are irrelevant to the question. Detailed case is shown in Fig. 6. This phenomenon is particularly obvious in the multi-hop reasoning process of complex questions.

Therefore, logic drift specifically focuses on the specific hallucination phenomenon where the LLM output is logically inconsistent with both the question intent and the knowledge graph. This phenomenon is particularly prominent in complex Knowledge Graph Question Answering tasks.

## M  LIMITATIONS OF OUR WORK

To the best of our knowledge, our method primarily contains the following limitation:

Due to the excessively large search space for correct reasoning paths, although our method's predicted candidate paths contain logic-consistent correct reasoning paths, they still inevitably introduce a certain number of incorrect reasoning paths. This can be attributed to the inherent limitations of the beam search strategy we adopted.

## N  BROADER IMPACT OF OUR WORK

Our work focuses on enhancing the logical reasoning capabilities of LLMs in structured knowledge reasoning from an output perspective, enabling them to maintain logical consistency and achieve precise reasoning. The positive impact of our work is to provide the knowledge graph reasoning and natural language processing communities with a flexible and transferable logic-consistency reasoning framework, offering important technical support for building more trustworthy and interpretable artificial intelligence systems. We do not believe our method has any negative societal impact, and we will endeavor to prevent the misuse of our approach.

## O  THE USE OF LARGE LANGUAGE MODELS

We declare that Large Language Models (LLMs) were used solely for language polishing and grammatical corrections to improve the writing quality of this work. No LLMs were involved in the generation of research ideas, methodology, experimental design, data analysis, core content creation, or retrieval and discovery of information.

