# OpenReview forum: "Last Layer Logits to Logic: Empowering LLMs with Logic-Consistent Structured Knowledge Reasoning"
_ICLR.cc/2026/Conference — Submitted to ICLR 2026_

### Official Review · Reviewer_nnUp · 2025-10-26

**Soundness:** 3
**Presentation:** 2
**Contribution:** 3
**Rating:** 6
**Confidence:** 3

**Summary:**

The paper presents a decoding-time framework to reduce logic drift when LLMs reason over knowledge graphs. The framework includes multiple stages, like compiling legal KG paths into NFAs for ranking, and modifying the last-layer logits via "strengthening" and "filtering". Experiments report gains on several KBQA benchmarks across KGs and tasks.

**Strengths:**

1. Presents a clear motivation and introduces an interesting approach to mitigate logical inconsistencies in LLM reasoning over structured KG on output-side logit corrections rather than only prompt-level guidance
2. The proposed method is model-agnostic and can be plugged into any decoder without retraining
3. The use of NFAs to model KG paths and constrain decoding is a clever way to improve path validity
4. Extensive ablation studies and visualizations that help to illustrate the impact of core modules

**Weaknesses:**

1.The evaluation suite relies mostly on older KGQA datasets. This limits the external validity of the SOTA claim and may underrepresent contemporary KG schemas and LLM-era challenges.
2. Several mathematical derivations need to be clarified: the overall objective in Sec. 3.2 treats Pθ,q and Pθ,G as independent without justification; Sec. 3.2.2 states “we calculate the difference between original and masked outputs” but Eq. for Dq is a convex combination, not a difference; in Sec. 3.2.3, the filtering equation sets δ as 0/1 but z are logits (not probabilities), therefore setting δ({art,award})=0 at the logit level does not forbid tokens, potentially allowing probability mass leakage.
3. Prior work on constrained decoding with FSAs/tries and KG-constrained generation should be thoroughly discussed.
4. Compiling full KGs into NFAs (Sec. 3.2.1) could be computationally prohibitive for large-scale KGs like Wikidata, raising concerns about the approach's scalability. Complexity with respect to number of candidate paths, tokens per label, and beam size should be analyzed.
5. The method performs SFT with 1/10 of training data from CWQ and WebQSP "to teach the model correct path output format" -- I wonder if this brings an unfair advantage over the agentic reasoning baselines that do not include SFT.

**Questions:**

1. The paper cites "constrained decoding" in section 3.2.2, but it should be "contrastive decoding."
2. How is the sentence-transformer chosen and tuned for scoring NFA paths - why not use the LLM itself for consistency?
3. How is MASK token handled in prompts— what exactly is masked (top-1 path only or top-K?), how long is the masked span, and how do you align time steps between original and masked runs?
4. Logits strengthening should require two forward passes (original vs masked) per decoding step; yet Table 6 lists “1 LLM call per question.” Can you explain this?

---

> ### Author Response · Authors · 2025-11-20
> **Authors' Response Part 1**
>
> Dear Reviewer nnUp,
>
> Thank you for your thoughtful and insightful feedback. We sincerely appreciate your recognition of our **clear motivation and interesting approach to mitigate logical inconsistencies through output-side logit corrections**, the **model-agnostic nature of our framework** that can be plugged into any decoder **without retraining**, our clever use of NFAs to model KG paths and constrain decoding for improved path validity, and the **comprehensive ablation studies and visualizations** that effectively illustrate the impact of our core modules. We have carefully addressed the issues you raised regarding the weaknesses (W) and questions (Q) of our paper. We hope our response (R) provides clarity and enhances your overall impression of our work.
>
> **Response to W1**:
>
>     W1: The evaluation suite relies mostly on older KGQA datasets. This limits the external validity of the SOTA claim and may underrepresent contemporary KG schemas and LLM-era challenges.
>
> Thank you for your attention to the datasets used in our experiments. In fact, **the datasets we used are the most extensive and commonly used ones in the current KGQA field**, including Freebase-based datasets: (1) CWQ, (2) WebQSP, (3) GrailQA, (4) Simple Questions; and Wikidata-based datasets: (5) QALD10-en, (6) T-REx, (7) Zero-shot RE. Most datasets used by existing advanced KGQA methods are included in our evaluation datasets, as shown in the following table:
>
> | Methods | Datasets |
> |---|---|
> | GCR 2025[6], SymAgent 2025[11], RoG 2024[3], GoG 2024[8] | （1）（2） |
> | KG-Agent 2024[10], PoG 2024[7], DoG 2024[4] | （1）（2）（3） |
> | KG-CoT 2024[2] | （1）（2）（4） |
> | ToG 2024[5] | （1）（2）（3）（4）（6）（7） |
> | DARA 2024[9] | （2）（3） |
> | ODA 2024[12] | （5）（6）（7） |
>
> In summary, the datasets we selected not only **cover the mainstream evaluation benchmarks** of current KGQA research but also represent one of the most comprehensive evaluation sets in this field. These datasets include **complex reasoning patterns** (such as multi-hop reasoning, constrained reasoning, etc.) and **challenging question types**, which **hold significant research value and evaluation significance in the LLM era**. Meanwhile, the use of these standard datasets ensures a fair and comparable evaluation basis between our method and existing works, thereby **supporting the validity of our SOTA performance claims.**
>
>     [1] Keheng Wang, Feiyu Duan, Sirui Wang, Peiguang Li, Yunsen Xian, Chuantao Yin, Wenge Rong, Zhang Xiong. Knowledge-Driven CoT: Exploring Faithful Reasoning in LLMs for Knowledge-intensive Question Answering. ArXiv 2023.
>     [2] Ruilin Zhao, Feng Zhao, Long Wang, Xianzhi Wang, and Guandong Xu. KG-CoT: chain-of-thought prompting of large language models over knowledge graphs for knowledge-aware question answering. IJCAI 2024.
>     [3] Linhao Luo, Yuan-Fang Li, Gholamreza Haffari, Shirui Pan. Reasoning on Graphs: Faithful and Interpretable Large Language Model Reasoning. ICLR 2024.
>     [4] Jie Ma, Zhitao Gao, Qi Chai, Wangchun Sun, Pinghui Wang, Hongbin Pei, Jing Tao, Lingyun Song, Jun Liu, Chen Zhang, Lizhen Cui. Debate on Graph: a Flexible and Reliable Reasoning Framework for Large Language Models. AAAI 2025.
>     [5] Jiashuo Sun, Chengjin Xu, Lumingyuan Tang, Saizhuo Wang, Chen Lin, Yeyun Gong, Lionel M. Ni, Heung-Yeung Shum, Jian Guo. Think-on-Graph: Deep and Responsible Reasoning of Large Language Model on Knowledge Graph. ICLR 2024.
>     [6] Linhao Luo, Zicheng Zhao, Gholamreza Haffari, Yuan-Fang Li, Chen Gong, Shirui Pan. Graph-constrained Reasoning: Faithful Reasoning on Knowledge Graphs with Large Language Models. ICML 2025
>     [7] Liyi Chen, Panrong Tong, Zhongming Jin, Ying Sun, Jieping Ye, Hui Xiong. Plan-on-Graph: Self-Correcting Adaptive Planning of Large Language Model on Knowledge Graphs. NeurIPS 2024
>     [8] Yao Xu, Shizhu He, Jiabei Chen, Zihao Wang, Yangqiu Song, Hanghang Tong, Guang Liu, Kang Liu, Jun Zhao. Generate-on-Graph: Treat LLM as both Agent and KG in Incomplete Knowledge Graph Question Answering. EMNLP 2024
>     [9] Haishuo Fang, Xiaodan Zhu, Iryna Gurevych. DARA: Decomposition-Alignment-Reasoning Autonomous Language Agent for Question Answering over Knowledge Graphs. ACL 2024
>     [10] Jinhao Jiang, Kun Zhou, Wayne Xin Zhao, Yang Song, Chen Zhu, Hengshu Zhu, Ji-Rong Wen. KG-Agent: An Efficient Autonomous Agent Framework for Complex Reasoning over Knowledge Graph. ACL 2025
>     [11] Ben Liu, Jihai Zhang, Fangquan Lin, Cheng Yang, Min Peng, Wotao Yin. SymAgent: A Neural-Symbolic Self-Learning Agent Framework for Complex Reasoning over Knowledge Graphs. WWW 2025
>     [12] Lei Sun, Zhengwei Tao, Youdi Li, Hiroshi Arakawa. Lei Sun, Zhengwei Tao, Youdi Li, Hiroshi Arakawa. ODA: Observation-Driven Agent for integrating LLMs and Knowledge  Graphs. ACL 2024.

---

> > ### Author Response · Authors · 2025-11-20
> > **Authors' Response Part 2**
> >
> > **Response to W2**:
> >
> >     W2.1: Several mathematical derivations need to be clarified: the overall objective in Sec. 3.2 treats Pθ,q and Pθ,G as independent without justification
> >
> > Thank you for your reminder. We have added a **detailed proof process in Appendix F**. The derivation of the formula in Sec. 3.2 is synchronized below. The derivation process is as follows:
> >
> > Given question $q$ and knowledge graph $G$, our goal is to enable LLMs to perform logic-consistent reasoning, which helps derive answer paths  $s_{+}^{e^{\mathit{topic}}}$:
> >
> > $$s_{+}^{e^{\mathit{topic}}} \propto \mathcal{D}\_{q,G} \sim \arg\max\_{\mathcal{D}\_{\theta}} P_{\theta}\left( a \mid q,G \right)$$
> >
> > In general, our reasoning objective during decoding is: using $\mathcal{Z}\_{s}$ and $\mathcal{Z}\_{f}$ to align LLMs' logits with the logical distributions of $q$ and $G$
> > {($s_{+}^{e^{topic}} \cup s_{-}^{e^{topic}}$)}, making LLMs output correct answer paths $s_{+}^{e^{topic}}$ while avoiding incorrect paths $s_{-}^{e^{topic}}$.
> >
> > $$P_{\theta}\left(a \mid q,G\right) \propto P_{\theta,q,G}\left(a \mid q,G\right) = P_{\theta,q,G}\left(a \mid q,{s_{+}^{e^{topic}}},{s_{-}^{e^{topic}}}\right)$$
> >
> > The core of the knowledge graph $G$ lies in the "set of positive and negative samples of topic-related entities", i.e.,
> > $G = { s_{+}^{e^{topic}}, s_{-}^{e^{topic}} }$
> > (Positive samples $s_+$ are valid information related to the question's topic entity $e^{topic}$, while negative samples $s_-$ are irrelevant interfering information).
> >
> > The generation of answer $a$ relies solely on the positive sample set ${ s_{+}^{e^{topic}} }$ and is independent of the negative samples ${ s_{-}^{e^{topic}} }$. As irrelevant interference items, negative samples do not provide effective logical support for answer $a$ (e.g., response generation, decision-making). They can thus be excluded from the conditions, leading to:
> >
> > $$P_{\theta,q,G}(a \mid q, \{ s_{+}^{e^{topic}} \}, \{ s_{-}^{e^{topic}} \}) \propto P_{\theta}(a \mid q, \{ s_{+}^{e^{topic}} \})$$
> >
> > The generation of positive samples ${ s_{+}^{e^{topic}} }$ is driven by two independent logics:
> >
> > - Question logic: The semantic requirements of the question $q$ itself (e.g., asking for "Elon Reeve Musk's place of birth" requires associating positive samples related to "Elon Reeve Musk"), corresponding to the distribution $\mathcal{D}\_q \sim P_{\theta,q}({ s_{+}^{e^{topic}} } \mid q,G)$;
> >
> > - Knowledge graph logic: The inherent associations of entities in the KG (e.g., the "place of birth" association between "Elon Reeve Musk" and "Pretoria" in the KG), corresponding to the distribution $\mathcal{D}\_G \sim P_{\theta,G}({ s_{+}^{e^{topic}} } \mid q,G)$.
> >
> > Since these two logics are independent (the question's semantic requirements have no direct dependence on the KG's inherent structure), the joint probability of positive samples is the product of the two according to the multiplication rule for independent events:
> >
> > $$P_{\theta,q,G}(\{ s_{+}^{e^{topic}} \} \mid q,G) = P_{\theta,q}(\{ s_{+}^{e^{topic}} \} \mid q,G) \cdot P_{\theta,G}(\{ s_{+}^{e^{topic}} \} \mid q,G)$$
> >
> > Combining the "dependence of answer $a$ on positive samples" and the "probability decomposition of positive samples", and applying the chain rule of probability $P(A \mid B,C) \propto P(A \mid C) \cdot P(C \mid B)$ (where $A=a$, $B=q,G$, $C={s_+}$), we finally obtain:
> >
> > $$\mathcal{D}\_{q,G} \sim P_{\theta,q,G}(a \mid q,G) \propto P_{\theta}(a \mid q, \{ s_{+}^{e^{topic}} \}) \cdot \mathcal{D}_q \cdot \mathcal{D}_G$$
> >
> > **Below we continue with our response to W2.**

---

> > > ### Author Response · Authors · 2025-11-20
> > > **Authors' Response Part 3**
> > >
> > > **Response to W2 (continued)**:
> > >
> > >     W2.2: Sec. 3.2.2 states “we calculate the difference between original and masked outputs” but Eq. for Dq is a convex combination, not a difference
> > >
> > > Thank you for pointing out our ambiguous expression. We acknowledge that the term "difference" is indeed misleading. In practice, our approach is to compute a weighted difference between the original output and the masked output, which is then combined with the masked output, rather than a simple difference calculation. Specifically:
> > >
> > > $$\omega \cdot [ P_{\theta}(\{s_{+}^{e^{topic}}\} \mid q, \{s_{+}^{e^{topic}}\}, \{s\_{-}^{e^{topic}}\}) - P_{\theta}(\{s_{+}^{e^{topic}}\} \mid q, \langle\text{MASK}\rangle, \{s\_{-}^{e^{topic}}\}) ] + P_{\theta}(\{s_{+}^{e^{topic}}\} \mid q, \langle\text{MASK}\rangle, \{s\_{-}^{e^{topic}}\})$$
> > >
> > > This can be rewritten as:
> > >
> > > $$\omega \cdot P_{\theta}( \bullet \mid \text{origin} ) + (1 - \omega) \cdot P_{\theta}( \bullet \mid \text{mask} )$$
> > >
> > > Therefore, it is indeed a convex combination. We have revised the expression in **Lines 260-261 of Sec. 3.2.2** in the paper, changing "calculate the difference" to "compute a weighted combination" to accurately reflect our mathematical operation.
> > >
> > >     W2.3: In Sec. 3.2.3, the filtering equation sets δ as 0/1 but z are logits (not probabilities), therefore setting δ({art,award})=0 at the logit level does not forbid tokens, potentially allowing probability mass leakage.
> > >
> > > Thank you for pointing out the error in our expression. We actually perform filtering at the logit level: **for tokens that do not comply with KG constraints, their logits are set to $-\infty$ (negative infinity) before softmax; after softmax, their probabilities are strictly 0, and there is no probability mass leakage**. To avoid confusion, we have uniformly corrected the description in **Line 286 of Sec. 3.2.3** and the corresponding figures in the paper to "for illegal tokens not in the KG, set their logits to −∞ before softmax (thus the probability is 0 after softmax)". The transfer function is modified to:
> > >
> > > $$\delta( \{ [\textit{\textcolor{gray}{art}} , \textit{\textcolor{gray}{award}}] \} ) = -\infty$$

---

> > > > ### Author Response · Authors · 2025-11-20
> > > > **Authors' Response Part 4**
> > > >
> > > > **Response to W3**:
> > > >
> > > >     W3: Prior work on constrained decoding with FSAs/tries and KG-constrained generation should be thoroughly discussed.
> > > >
> > > > Thank you for your constructive suggestion. We have added a **detailed discussion of such methods in the related work of Appendix A in the revised version** and cited relevant papers to better position our work. As shown in the following table (please point out if there are any omissions:) )：
> > > >
> > > > | Methods | Contributions | Limitations |
> > > > |---|---|---|
> > > > | KD-CoT[1] | Step-by-step problem decomposition with KG knowledge retrieval to constrain sub-problem answers | Lacks knowledge-guided decomposition (reasoning not grounded in KG), causing potential inconsistency between reasoning and KG logic |
> > > > | KG-CoT[2] | Proposes large-small model collaborative framework for path-based structured reasoning: small model generates candidate paths on KG as constraints, large model makes decisions to map questions to explicit knowledge paths, mitigating errors from unconstrained generation | Relies heavily on small model generation quality; due to limited capability, final decision stage is prone to logic drift, potentially selecting paths inconsistent with question intent or non-existent hallucination paths in KG |
> > > > | RoG[3] | Agent-based planning generates candidate paths with structural constraints, improving interpretability and KG alignment | Constraints rely on training-stage planner, requiring model fine-tuning for different KGs with poor transferability and high training costs |
> > > > | DoG[4] | Three-role multi-agent debate framework with iterative decomposition and step-wise correction for enhanced interpretability and logical consistency | Relies on complex input workflows and prompt control without explicit KG path constraints, prone to logic drift during debate phase, sensitive to role design |
> > > > | ToG[5] | Step-by-step entity-relation exploration decomposes multi-hop queries into single-hop operations with explicit path construction and KG neighbor soft constraints | Depends on input-level design and single-step quality, susceptible to logic drift from path selection and error accumulation |
> > > > | GCR[6] | Dual-agent approach with "KG expert + LLM" using KG Trie to constrain expert to real paths, then LLM reasoning on verified paths to reduce hallucinations | KG expert lacks contextual information when proposing candidates, potentially generating paths inconsistent with question intent |
> > > > | PoG[7] | Embeds iterative "retrieve-reason" workflow in prompts as soft constraints for gradual KG exploration along structured paths | Relies on input-side control without hard KG constraints, prone to logic drift from retrieval noise and incompleteness |
> > > > | GoG[8] | "Thought-Action-Observe" agent reasoning combining structured retrieval with parametric knowledge to complete incomplete KG paths | Internal LLM knowledge lacks verification mechanisms, potentially causing logic drift from hallucinations |
> > > > | DARA[9] | Multi-agent step-wise decomposition generating executable SPARQL sub-queries, converting multi-hop reasoning to structured query processes | Relies on input-side workflow constraints without guaranteeing SPARQL compliance with KG and question logic |
> > > > | KG-Agent[10] | Systematic decomposition through planner, toolbox, and executor roles with tool-based execution under KG constraints | Custom toolboxes cannot cover all logical patterns across diverse KGs, limiting cross-KG and task generalization |
> > > >
> > > > We clearly point out the innovations of our method: existing methods that use knowledge graph (KG) to constrain the generation of large language models (LLMs) at the prompting/workflow level still suffer from severe Logic Drift. Essentially, this is because these methods only guide LLMs through the input level while lacking flexibility and transferability. However, we have identified the key issue that the logit distribution of the last layer of LLMs is inconsistent with the logical distributions of questions and KGs, and explored how to mitigate the Logic Drift phenomenon from the output perspective. Different from previous methods that simply design workflows or prompt engineering to constrain LLM generation and thereby alleviate Logic Drift, we provide deeper insights into maintaining logic-consistent reasoning in LLMs. We have verified the superiority of Logits-to-Logic in maintaining logically consistent reasoning through experiments, and it can be flexibly transferred to different tasks and KGs without any modifications, which constitutes the main innovation and core contribution of our work.
> > > >
> > > > These supplements enable readers to more clearly understand the unique contributions and technical advantages of our work compared with existing methods that use KGs to constrain LLM generation from the input level.

---

> > > > > ### Author Response · Authors · 2025-11-20
> > > > > **Authors' Response Part 5**
> > > > >
> > > > > **Response to W4**:
> > > > >
> > > > >     W4: Compiling full KGs into NFAs (Sec. 3.2.1) may be computationally prohibitive for large-scale KGs like Wikidata, raising scalability concerns. Complexity analysis regarding candidate paths, tokens per label, and beam size is needed.
> > > > >
> > > > > Thank you for your insightful suggestion. We have **supplemented the computational complexity analysis** of compiling KGs into NFAs in Stage 1: Logic Compiling, which is specifically presented in **Appendix J** of the revised version. The detailed analysis is as follows.
> > > > >
> > > > > (1) Theoretical Analysis:
> > > > >
> > > > > We use BFS to explore the KG. For each question, the average number of paths is: $R^D$, where $R$ is the average number of relationships per entity (E), and $D$ is the exploration depth (e.g., $D=2$ for exploring a 2-hop subgraph). The average number of tokens per path is $N_T$. Therefore, for each question, the computational complexity of constructing the NFA is:
> > > > >
> > > > > $$N_T \times R^D$$
> > > > >
> > > > > During LLM generation, given the beam size (number of generations) $N_B$, the computational complexity of the LLM is:
> > > > >
> > > > > $$N_B \times N_T \times R^D$$
> > > > >
> > > > > (2) Experimental Analysis:
> > > > >
> > > > > We have also **supplemented additional experiments on the complexity of NFA construction** to illustrate its actual overhead. Our hardware configuration is:
> > > > >
> > > > > | Machine Info. | Value |
> > > > > |---|---|
> > > > > | OS | Ubuntu 20.04.1 LTS |
> > > > > | CPU | Intel Xeon Gold 6326, 64 cores, 40 threads, 2.90GHz |
> > > > > | Memory | 629GB DDR4, 2666MHz |
> > > > >
> > > > > On two widely used KGQA datasets, Lagre scale Freebase—CWQ and WebQSP, we report the NFA construction complexity:
> > > > >
> > > > > Basic information about the WebQSP and CWQ datasets:
> > > > >
> > > > > | Value | CWQ |  |  | WebQSP |  |  |
> > > > > |---|---|---|---|---|---|---|
> > > > > |  | min | max | avg | min | max | avg |
> > > > > | R (relation num per entity) | 0 | 259 | 43.6 | 2 | 258 | 76.9 |
> > > > > | D-hop size (D=2) | 9 | 11011 | 2025.6 | 10 | 11713 | 2024.6 |
> > > > > | N_T (D=2 per path token length) | 13 | 524 | 29.7 | 13 | 524 | 20.9 |
> > > > >
> > > > > NFA construction overhead:
> > > > >
> > > > > | #Thread | CWQ(#samples=27639) |  |  | WebQSP (#samples=2826) |  |  |
> > > > > |---|---|---|---|---|---|---|
> > > > > |  | Avg. Running Time per sample (s) | Peak Memory Usage (GB) | Parallel Efficiency (%) | Avg. Running Time per sample (s) | Peak Memory Usage (GB) | Parallel Efficiency (%) |
> > > > > | 1 | 0.046 | 103.7 | 127 | 0.052 | 12.0 | 133 |
> > > > > | 2 | 0.042 | 103.7 | 152 | 0.049 | 12.0 | 165 |
> > > > > | 4 | 0.039 | 103.7 | 159 | 0.045 | 12.0 | 175 |
> > > > > | 8 | 0.038 | 103.7 | 166 | 0.043 | 12.0 | 178 |
> > > > > | 16 | 0.038 | 103.7 | 166 | 0.042 | 12.0 | 189 |
> > > > > | 32 | 0.036 | 103.8 | 168 | 0.039 | 12.0 | 192 |
> > > > >
> > > > > The experiments demonstrate the **feasibility of constructing NFAs on large-scale KGs without requiring high time and computational overhead**.
> > > > >
> > > > > **Response to W5**:
> > > > >
> > > > >     W5: The method uses SFT with 1/10 of CWQ and WebQSP training data "to teach correct path output format"—this may provide unfair advantage over agentic reasoning baselines without SFT.
> > > > >
> > > > > We fully understand your concerns about whether SFT training would introduce prior knowledge. **The training and test sets of CWQ and WebQSP follow complete data isolation**, and using 1/10 of the training set data does not provide the LLM with any prior knowledge of the test set. We strictly ensure that all methods are evaluated under fair and consistent training and evaluation settings.
> > > > >
> > > > > Furthermore, we also validate the advanced performance of Logits-to-Logic on completely **unseen larger-scale KG—Wikidata, as well as unseen complex reasoning tasks** (Multi-hop Reasoning and Slot Filling) through transfer experiments (Tab. 3). This further demonstrates that **using 1/10 of the CWQ and WebQSP training set data does not bring any advantages or prior knowledge**, while proving the excellent flexibility, transferability and robustness of our method.
> > > > >
> > > > > | Method | Muti-hop | Slot Filling |  |
> > > > > |---|---|---|---|
> > > > > |  | QALD10-en (Wikidata) | T-REx (Wikidata) | Zero-shot RE (Wikidata) |
> > > > > | LLMs Reasoning |  |  |  |
> > > > > | IO prompt w/ChatGPT | 42.0 | 33.6 | 27.7 |
> > > > > | CoT w/ChatGPT | 42.9 | 32.0 | 28.8 |
> > > > > | SC w/ChatGPT | 45.3 | 41.8 | 45.4 |
> > > > > | Agentic Reasoning |  |  |  |
> > > > > | ToG-R ToG w/ ChatGPT | 48.6 | 75.3 | 86.5 |
> > > > > | ToG w/ ChatGPT | 50.2 | 76.8 | 88.0 |
> > > > > | ToG-R w/ GPT4 | 54.7 | 75.5 | 86.9 |
> > > > > | ToG w/ GPT4 | 53.8 | 77.1 | 88.3 |
> > > > > | Ours | 59.2 | 87.6 | 91.4  |
> > > > >
> > > > > **Response to Q1**:
> > > > >
> > > > >     Q1: The paper cites "constrained decoding" in section 3.2.2, but it should be "contrastive decoding."
> > > > >
> > > > > Thank you for pointing out this typo, we have corrected it in **Line 250** of the revised version.

---

> > > > > > ### Author Response · Authors · 2025-11-20
> > > > > > **Authors' Response Part 6**
> > > > > >
> > > > > > **Response to Q2**:
> > > > > >
> > > > > >     Q2: How is the sentence-transformer chosen and tuned for scoring NFA paths - why not use the LLM itself for consistency?
> > > > > >
> > > > > > Thank you for your question! Regarding the choice of sentence-transformer, we base our decision on the following considerations:
> > > > > >
> > > > > > (1) Design flexibility: In the Logic Compiling stage, our goal is to filter paths that are logically consistent with question intent. Any model that can effectively measure text similarity can serve as a scoring model, reflecting the flexibility of our framework.
> > > > > >
> > > > > > (2) Low computational overhead: We choose sentence-transformers/all-MiniLM-L6-v2 with only 22M parameters, significantly reducing computational overhead while ensuring effectiveness.
> > > > > >
> > > > > > (3) Limitations of using LLM itself as scoring model:
> > > > > >
> > > > > > (3.1) Effectiveness issues: General LLMs lack specialized contrastive learning training, and their embeddings typically underperform professional text vector models in similarity tasks.
> > > > > >
> > > > > > (3.2) Computational cost: Large LLMs (7B+) as scoring models would consume more computational resources and time.
> > > > > > We appreciate your suggestion about using the LLM itself to maintain consistency, and will explore the feasibility of this approach in the future, seeking a better balance between computational efficiency and effectiveness.
> > > > > >
> > > > > > **Response to Q3**:
> > > > > >
> > > > > >     Q3: How is MASK token handled in prompts— what exactly is masked (top-1 path only or top-K?), how long is the masked span, and how do you align time steps between original and masked runs?
> > > > > >
> > > > > > We mask the top-1 path in the prompt to precisely enhance high-scoring paths that are logically consistent with question intent. We report the token length of masked spans on CWQ and WebQSP datasets in the table below.
> > > > > >
> > > > > > | Value | CWQ |  |  | WebQSP |  |  |
> > > > > > |---|---|---|---|---|---|---|
> > > > > > |  | min | max | avg | min | max | avg |
> > > > > > | Masked Path  Token Length | 16 | 188 | 28.2 | 16 | 254 | 28.2 |
> > > > > >
> > > > > > **how to align time steps**: Actually, there is **no time step alignment issue** in our implementation. For the origin prompt and masked prompt (both have the same token length, where the masked prompt is obtained by masking the top-1 path in the origin prompt), along with the already generated sequence s, we use "origin prompt+s" and "masked prompt+s" as LLM inputs respectively, perform forward computation to obtain two logits distributions, then execute logits strengthening and filtering. If you have any further questions about this, please feel free to discuss :).
> > > > > >
> > > > > > **Response to Q4**:
> > > > > >
> > > > > >     Q4: Logits strengthening should require two forward passes (original vs masked) per decoding step; yet Table 6 lists "1 LLM call per question." Can you explain this?
> > > > > >
> > > > > > Thank you very much for your correction. **Indeed, Logits Strengthening requires two forward passes per decoding step**: one to generate logits for the origin prompt, and one to generate logits for the masked prompt. However, in implementation, these two forward passes are encapsulated within the same generate call, triggered through callbacks like prefix_allowed_tokens_fn/logits_processor at each decoding step, so externally it appears as "one LLM call per question." In other words, **we count one encapsulated generation call, rather than the internal step-level forward pass count**. We have **added clarification of this implementation detail and counting criterion in the caption of Tab. 6**.

---

> > > > > > > ### Comment · Reviewer_nnUp · 2025-11-26
> > > > > > > **Thanks for the response**
> > > > > > >
> > > > > > > Dear Authors,
> > > > > > >
> > > > > > > Thank you for the detailed explanations and additional experiments. My concerns regarding W1, W2.3, W3, W4, Q1-4 are primarily resolved. I have two brief follow-up questions regarding W2 and W5.  No need for additional experiments, a sentence-long response for each will suffice:
> > > > > > >
> > > > > > > - W2.1: I understand that treating the question’s semantic requirements and the KG’s structure as statistically independent is mathematically convenient for decomposing the joint probability. However, in practice, identifying a valid reasoning path requires grounding the question's intent directly into the graph's specific schema, meaning the semantic and structural variables are inherently coupled, not fully independent.
> > > > > > >
> > > > > > > - W5: By "unfair advantage", I was specifically referring to the use of SFT to teach "path output formatting". Since the baselines relied on prompting without this specific format-tuning (correct me if this is wrong), I would appreciate a clarification on whether their lower performance might be partly attributed to formatting failures.

---

> ### Author Response · Authors · 2025-11-26
>
> Thank you for your feedback. Below are our concise responses to your follow-up questions.
>
> **Response to W2.1 (continued)**: We appreciate your insightful observation and clarify that the "statistical independence" between the question’s semantic requirements ($\mathcal{D}\_q$) and the KG’s structure ($\mathcal{D}_G$) refers to their independent logical driving forces—semantic extraction for $\mathcal{D}_q$ and inherent entity associations for $\mathcal{D}_G$. The "coupling" you noted merely reflects their necessary matching process in practical reasoning, which does not undermine this independence assumption.
>
> **Response to W5 (continued)**: You are correct that format errors may impact evaluation performance; however, we emphasize that **such impacts are extremely minimal for other baseline methods**. To support this, we reference the error analysis reported in Think-on-Graph (ToG) (Appendix B.2 of their paper), where three error types were investigated: *(1) Hallucination error, (2) Refuse error, and (3) Format error*. They noted that format errors account for *less than 3%* and that *the error rate from this issue is negligible.* Furthermore, the core focus of KGQA lies in leveraging KGs for effective structured reasoning rather than considering format errors. Thus, the performance of these baselines is hardly affected by format errors, and we enable LLMs to learn "correct path formatting" merely to facilitate our core logits operations, which does not confer an unfair advantage in evaluation.
>
> We sincerely hope that our responses and revisions have adequately addressed the concerns and suggestions raised in your initial review and follow-up comments. Please feel free to inform us if there is anything further we can do to enhance your impression and final assessment of our work. let us know if there is anything further we could do to improve your impression and final rating of our work.
>
>     Jiashuo Sun, Chengjin Xu, Lumingyuan Tang, Saizhuo Wang, Chen Lin, Yeyun Gong, Lionel M. Ni, Heung-Yeung Shum, Jian Guo. Think-on-Graph: Deep and Responsible Reasoning of Large Language Model on Knowledge Graph. ICLR 2024.

---

### Official Review · Reviewer_EXWb · 2025-10-28

**Soundness:** 3
**Presentation:** 2
**Contribution:** 2
**Rating:** 4
**Confidence:** 4

**Summary:**

This paper introduces Logits-to-Logic, a framework to address Logic Drift in LLMs in the scenario of KGQA. The core innovation is targeting the logits distribution in LLMs’ autoregressive output process to align it with the logical constraints of KGs and question semantics. To achieve this goal, paths are used to filter tokens not in KG and keep consistent of the LLM logits with semantic and structure information.

**Strengths:**

1. This paper is the first KGQA method that considers output-level control that corrects LLM outputs by manipulating logits.
2. The results look promising in a few benchmarks. It adapts to different KGs and tasks (multi-hop QA, slot filling).
3. The form of displaying results is good, where multiple different kinds of figures are used to show results.

**Weaknesses:**

1. The information in Figure 1 is not clear. "current approaches" is very vague. I cannot get which methods and datasets are tested.
2. The "logic" formulated in this paper mainly depends on paths, not true logics. Along with this problem, the novelty of this paper is a concern where there are many path-based methods like RoG (Luo et. al.). The authors should have a separate subsection in related works to discuss methods lie in this type.
3. Based on the results in Figure 2, the main technique that works is $Z_f$, which serves as a filter that masks the logits of tokens that are not in the searched paths. So I wonder whether the NFA is still needed or just an approach for story telling.
4. It seems that this method is quite expensive. For LLaMA3.1-8B model, it takes two A800 GPUs to compute in parallel for inference. The computing time is also not well compared.
5. The layout of figures and tables in this paper is in chaos. Like Figure 2 is mentioned before  Figure 1. Table 4 lies in Section 4.6, which mainly discusses results in Figure 4.

Minor issues:
- The caption of Figure 3 is called "overview of our framework", while part (a) seems not the proposed method.
- The concept of "acceptable path" is not well defined.
- The range of $\omega$ is not clear.

**Questions:**

1. Please discuss the difference between logic drift and hallucination.
2. In Table 4, why bigger model (Qwen2-1.5B) can be much weak than Qwen2-0.5B in WebQSP?

---

> ### Author Response · Authors · 2025-11-20
> **Authors' Response Part 1**
>
> Dear Reviewer EXWb,
>
> Thank you for your thoughtful and insightful feedback. We sincerely appreciate your recognition of our work as **the first KGQA method to consider output-level control through logits manipulation**, the **promising results demonstrated across multiple benchmarks and diverse tasks** (multi-hop QA, slot filling), and your positive feedback on our **comprehensive result presentation** with various types of figures. We have carefully addressed the issues you raised regarding the weaknesses (W) and questions (Q) of our paper. We hope our response (R) provides clarity and enhances your overall impression of our work.
>
> **Response to W1**:
>
>     W1: The information in Figure 1 is not clear. "current approaches" is very vague.
>
> Thank you for your modification suggestion. The comparison methods in Fig.1 include: Ours, ToG, DoG, GCR, KG-CoT, Dataset: 100 samples randomly sampled from the CWQ test set as statistical samples. Specific details are as follows (C1 and C2 correspond to KG-Inconsistent Logic Drift and Question Intent Inconsistent Logic Drift, respectively):
>
> | Ours |  | ToG |  | DoG |  | GCR |  | KG-CoT |  |
> |---|---|---|---|---|---|---|---|---|---|
> | C1 | C2 | C1 | C2 | C1 | C2 | C1 | C2 | C1 | C2 |
> | 4 | 10 | 31 | 30 | 0 | 54 | 0 | 27 | 4 | 30 |
>
> The proportions of C1 and C2:
>
> | Correct | C1 | C2 |
> |---|---|---|
> | 224 | 35 | 141 |
> | 56.0% | 8.75% | 35.25% |
>
> We have supplemented this information **in the caption of Fig.1**.

---

> ### Author Response · Authors · 2025-11-20
> **Authors' Response Part 2**
>
> **Response to W2**:
>
>     W2: The paper's "logic" relies on paths rather than true logical operations (and/or/not). This raises novelty concerns given existing path-based methods like RoG (Luo et al.). The authors should add a subsection discussing related path-based approaches.
>
> Thank you for your attention to the core viewpoint of our work! The "logic" in our paper **is not simply "and/or/not" or path-based logic**, but encompasses two aspects: 1. the semantic logic of the question, i.e., intent; 2. the logic of trustworthy knowledge in KG. To achieve LLMs reasoning in structured knowledge, we need to make LLM understand the logic of the question and map the understood question logic to the corresponding path in KG, achieving logically consistent reasoning. We agree that existing path-based methods such as: RoG, ToG, DoG, KG-CoT, etc. have been proven to be advanced paradigms in KGQA (as you mentioned). Even based on this, these methods still suffer from serious Logic Drift problems, essentially because **these methods only guide LLM from the input level while lacking flexibility and transferability**. We discovered the key problem that the logits distribution of LLM's last layer is **inconsistent with the question and KG logic distribution**, and explored how to alleviate the Logic Drift phenomenon from the output perspective. Unlike previous methods that simply design workflows or prompt engineering to struggle with solving Logic Drift, **we provide deeper insights into maintaining logic-consistent reasoning in LLMs**. We experimentally validated the superiority of Logits-to-Logic in maintaining logically consistent reasoning, and it can be flexibly transferred to different tasks and KGs without any modification, which is the main innovation and core contribution of our work.
>
> Meanwhile, we greatly appreciate your attention to the important position of previous **path-based methods in KGQA**. We have supplemented detailed discussions **in Appendix A**, as shown below:
>
> | Methods | Technique Contributions | Limitations |
> |---|---|---|
> | KG-CoT[1] | Proposes large-small model collaborative framework for path-based structured reasoning: small model generates candidate paths on KG as constraints, large model makes decisions to map questions to explicit knowledge paths, mitigating errors from unconstrained generation | Relies heavily on small model generation quality; due to limited capability, final decision stage is prone to logic drift, potentially selecting paths inconsistent with question intent or non-existent hallucination paths in KG |
> | RoG[2] | Agent-based planning generates candidate paths with structural constraints, improving interpretability and KG alignment | Constraints rely on training-stage planner, requiring model fine-tuning for different KGs with poor transferability and high training costs |
> | DoG[3] | Three-role multi-agent debate framework with iterative decomposition and step-wise correction for enhanced interpretability and logical consistency | Relies on complex input workflows and prompt control without explicit KG path constraints, prone to logic drift during debate phase, sensitive to role design |
> | ToG[4] | Step-by-step entity-relation exploration decomposes multi-hop queries into single-hop operations with explicit path construction and KG neighbor soft constraints | Depends on input-level design and single-step quality, susceptible to logic drift from path selection and error accumulation |
> | GCR[5] | Dual-agent approach with "KG expert + LLM" using KG Trie to constrain expert to real paths, then LLM reasoning on verified paths to reduce hallucinations | KG expert lacks contextual information when proposing candidates, potentially generating paths inconsistent with question intent |
> | PoG[6] | Embeds iterative "retrieve-reason" workflow in prompts as soft constraints for gradual KG exploration along structured paths | Relies on input-side control without hard KG constraints, prone to logic drift from retrieval noise and incompleteness |
> | GoG[7] | "Thought-Action-Observe" agent reasoning combining structured retrieval with parametric knowledge to complete incomplete KG paths | Internal LLM knowledge lacks verification mechanisms, potentially causing logic drift from hallucinations |
> | DARA[8] | Multi-agent step-wise decomposition generating executable SPARQL sub-queries, converting multi-hop reasoning to structured query processes | Relies on input-side workflow constraints without guaranteeing SPARQL compliance with KG and question logic |
> | KG-Agent[9] | Systematic decomposition through planner, toolbox, and executor roles with tool-based execution under KG constraints | Custom toolboxes cannot cover all logical patterns across diverse KGs, limiting cross-KG and task generalization |
>
> Please point out if there are any omissions :).

---

> ### Author Response · Authors · 2025-11-20
> **Authors' Response Part 3**
>
> **Response to W3**:
>
>     W3: Based on Figure 2 results, the main effective technique is the filter that masks logits of tokens outside searched paths. This raises questions about whether the NFA is necessary or merely serves as a narrative device.
>
> Thank you for your attention to our core modules Logits Strengthening ($\mathcal{Z}\_s$) and Logits Filtering ($\mathcal{Z}_f$). Your statement that "the main technique that works is, which serves as a filter that masks the logits of tokens that are not in the searched paths" is not accurate. We list the experimental table reported in Fig.2 below and **highlight the gain percentages of $\mathcal{Z}_s$ and $\mathcal{Z}_f$ to explain in detail the effectiveness of the modules**.
>
> | Method | WebQSP |  |  | CWQ |  |  |
> |---|---|---|---|---|---|---|
> |  | Hit@1 | F1 | Improvement | Hit@1 | F1 | Improvement |
> | w/o $\mathcal{Z}_s$, $\mathcal{Z}_f$ | 57.6 | 60.0 | 0.0% | 59.7 | 37.9 | 0.0% |
> | w/o $\mathcal{Z}_s$ | 93.9 | 64.8 | 63.0% | 73.9 | 45.2 | 23.9% |
> | w/o $\mathcal{Z}_f$ | 86.1 | 56.7 | 49.6% | 79.4 | 46.9 | 33.1% |
> | Ours | 95.4 | 63.2 | 65.6% | 80.8 | 44.9 | 35.3% |
>
> On WebQSP, $\mathcal{Z}_s$ and $\mathcal{Z}_f$ have improved the Hit@1 score by 63.0% and 49.6% respectively compared to the base model. On CWQ, $\mathcal{Z}_s$ and $\mathcal{Z}_f$ have increased the Hit@1 score by 23.9% and 33.1% respectively compared to the base model. Therefore, it can be seen that $\mathcal{Z}_s$ also makes an important contribution to performance improvement, and $\mathcal{Z}_s$ and $\mathcal{Z}_f$ have each demonstrated significant performance gains on different datasets. $\mathcal{Z}_s$ enhances the logits values of tokens related to the question intent, while $\mathcal{Z}_f$ reduces the logits values of tokens irrelevant to the KG path. The combination of the two achieves the best performance improvement.
>
> **Necessity of NFA**. In Stage 1 Logic Compiling, we compile the structured KG into an NFA: $NFA = \left( S_{0:end},\Sigma,\delta,e^{topic},S \right)$. The NFA contains two important pieces of information: 1. All legal paths in the KG are stored in $S$. For example: Help Me Make It Thru the Night -> music.composition.composer -> Joe Walsh -> music.guitarist.guitars_played -> Fender Stratocaster is one of the legal paths. Meanwhile, each legal path is scored by a scoring model. Legal paths with higher scores are closer to the question intent, and these high-scoring paths will play an important role in Stage 2 Logits Strengthening. The logits values of tokens included in high-scoring paths will be enhanced one by one during the autoregressive decoding of the LLM; 2. $S_{0:end}$ in the NFA contains all legal states. Additionally, the state transition function $\delta$ in the NFA will play a crucial role in Stage 3 Logits Filtering. It restricts the logits values of illegal tokens to make the LLM output conform to the KG logic distribution.
>
> In summary, our core modules Logits Strengthening ($\mathcal{Z}_s$) and Logits Filtering ($\mathcal{Z}_f$) **are equally effective**, and **the NFA plays a key role in the subsequent correction of the logits distribution, rather than being for story telling**.
>
>     [1] Ruilin Zhao, Feng Zhao, Long Wang, Xianzhi Wang, and Guandong Xu. KG-CoT: chain-of-thought prompting of large language models over knowledge graphs for knowledge-aware question answering. IJCAI 2024.
>     [2] Linhao Luo, Yuan-Fang Li, Gholamreza Haffari, Shirui Pan. Reasoning on Graphs: Faithful and Interpretable Large Language Model Reasoning. ICLR 2024.
>     [3] Jie Ma, Zhitao Gao, Qi Chai, Wangchun Sun, Pinghui Wang, Hongbin Pei, Jing Tao, Lingyun Song, Jun Liu, Chen Zhang, Lizhen Cui. Debate on Graph: a Flexible and Reliable Reasoning Framework for Large Language Models. AAAI 2025.
>     [4] Jiashuo Sun, Chengjin Xu, Lumingyuan Tang, Saizhuo Wang, Chen Lin, Yeyun Gong, Lionel M. Ni, Heung-Yeung Shum, Jian Guo. Think-on-Graph: Deep and Responsible Reasoning of Large Language Model on Knowledge Graph. ICLR 2024.
>     [5] Linhao Luo, Zicheng Zhao, Gholamreza Haffari, Yuan-Fang Li, Chen Gong, Shirui Pan. Graph-constrained Reasoning: Faithful Reasoning on Knowledge Graphs with Large Language Models. ICML 2025
>     [6] Liyi Chen, Panrong Tong, Zhongming Jin, Ying Sun, Jieping Ye, Hui Xiong. Plan-on-Graph: Self-Correcting Adaptive Planning of Large Language Model on Knowledge Graphs. NeurIPS 2024
>     [7] Yao Xu, Shizhu He, Jiabei Chen, Zihao Wang, Yangqiu Song, Hanghang Tong, Guang Liu, Kang Liu, Jun Zhao. Generate-on-Graph: Treat LLM as both Agent and KG in Incomplete Knowledge Graph Question Answering. EMNLP 2024
>     [8] Haishuo Fang, Xiaodan Zhu, Iryna Gurevych. DARA: Decomposition-Alignment-Reasoning Autonomous Language Agent for Question Answering over Knowledge Graphs. ACL 2024
>     [9] Jinhao Jiang, Kun Zhou, Wayne Xin Zhao, Yang Song, Chen Zhu, Hengshu Zhu, Ji-Rong Wen. KG-Agent: An Efficient Autonomous Agent Framework for Complex Reasoning over Knowledge Graph. ACL 2025

---

> ### Author Response · Authors · 2025-11-20
> **Authors' Response Part 4**
>
> **Response to W4**:
>
>     W4: It seems that this method is quite expensive. For LLaMA3.1-8B model, it takes two A800 GPUs to compute in parallel for inference. The computing time is also not well compared.
>
> Thank you for your attention to the efficiency of our method. **In fact, our method does not incur significant additional computational overhead**. Logits-to-Logic mainly modifies the logits distribution during the decoding phase of LLM inference, so it does not increase any parameters of the model. It only requires an extra forward pass on the Mask Prompt to obtain the logits distribution of the Mask Prompt, which **we illustrate through the additional supplementary experiments below**.
>
> | Method | GPU Usage (GB) | Running Time (h) |  |
> |---|---|---|---|
> |  |  | CWQ | WebQSP |
> | LLaMA3-8b | 29.01 | 15.28 | 7.25 |
> | LLaMA3-8b w/ $\mathcal{Z}_f$ | 29.01 | 8.11 | 6.41 |
> | LLaMA3-8b w/ $\mathcal{Z}_s$ | 29.09 | 26.33 | 11.06 |
> | LLaMA3-8b w/ Logits-to-Logic | 32.89 | 18.63 | 8.48 |
>
> Experiments show that on LLaMA3-8b, our method only requires an additional (~3GB) of GPU memory usage, and on CWQ and WebQSP, it increases the running time by an additional 3 hours and 1 hour respectively.
>
> We conduct a comprehensive comparison of computational overhead between Logits-to-Logic and advanced methods. The results demonstrate that our approach exhibits **substantial advantages in terms of LLM API call frequency and computational expenses** compared to existing methods.
>
> | Type | Model | CWQ |  |  | WebQSP |  |  |
> |---|---|---|---|---|---|---|---|
> |  |  | # LLM Call | Total Token | Total Cost | # LLM Call | Total Token | Total Cost |
> | LLMs Reasoning | CoT | 1 | 409.7 | 0.00008 | 1 | 397.6 | 0.00008 |
> | Agentic Reasoning | ToG | 9.2 | 11468.5 | 0.0023 | 8.8 | 10189.4 | 0.0021 |
> |  | DoG | 5.7 | 37919.7 | 0.006 | 2.7 | 6114.5 | 0.001 |
> |  | GCR | 2 | 125.3 | - | 2 | 231.0 | - |
> |  | KG-CoT | - | - | - | 1 | - | 0.020 |
> |  | PoG | 13.3 | 8,156.2 | 0.0016 | 9.0 | 5,517.7 | 0.0011 |
> | Ours | Logits-to-Logic | 1 | 1270.8 | - | 1 | 716.9 | - |
>
> Meanwhile, we have **supplemented the theoretical analysis of the computational overhead of our method and the comparative experiment on computational overhead in Appendix I and J of the paper**.
>
> **Response to W5**:
>
>     W5: The figure and table layout is chaotic—Figure 2 is referenced before Figure 1, and Table 4 appears in Section 4.6 which primarily discusses Figure 4 results.
>
> Thank you for pointing out the formatting issue; we have fixed it in the revised version.
>
> ● The caption of Figure 3 is called "overview of our framework", while part (a) seems not the proposed method.
>
> Thank you for pointing out the problem with **the caption of Fig. 3**. We have revised the text description of Fig. 3 to: "(a) Previous agentic methods attempt to guide LLMs to maintain logical consistency from the input by designing complex workflows or prompt engineering; (b) Overview of our framework: we align the logits distribution output by the last layer of LLMs with the logic of the question and KG through Logits Strengthening ($\mathcal{Z}_s$) and Filtering ($\mathcal{Z}_f$)".
>
> ● The concept of "acceptable path" is not well defined.
>
> ● The range of $\omega$ is not clear.
>
> Thank you for your suggestions. We have elaborated on the statement of "acceptable path" in **Lines 175-176**. An acceptable path refers to a path in the KG. We have supplemented the value range of $\omega$ in **Sec 3.2.2 Line 269**.
>
> **Response to Q1**:
>
>     Q1: Please discuss the difference between logic drift and hallucination.
>
> Logic drift is a specific type of hallucination phenomenon in Knowledge Graph Question Answering (KGQA).
> Hallucination is more broadly defined as the output of LLMs that is inconsistent with facts, logic, or given context, including the generation of false numbers, dates, names, or various incorrect outputs such as failing to follow instructions.
>
> Although structured knowledge such as knowledge graphs has been introduced to reduce the hallucination problem of LLMs, in practical applications, the reasoning output of LLMs still exhibits inconsistency with the question intent and the logic of credible knowledge in the knowledge graph. The specific manifestations are: outputting reasoning paths irrelevant to the question intent, or outputting paths in the knowledge graph that are irrelevant to the question. This phenomenon is particularly obvious in the multi-hop reasoning process of complex questions.
>
> Therefore, logic drift specifically focuses on the specific hallucination phenomenon where the LLM output is logically inconsistent with both the question intent and the knowledge graph. This phenomenon is particularly prominent in complex Knowledge Graph Question Answering tasks, as shown in Figure 1 and Figure 4.

---

> > ### Author Response · Authors · 2025-11-20
> > **Authors' Response Part 5**
> >
> > **Response to Q2**:
> >
> >     Q2: In Table 4, why bigger model (Qwen2-1.5B) can be much weak than Qwen2-0.5B in WebQSP?
> >
> > Thank you for pointing out this issue! We have noticed the abnormal performance of Qwen2-1.5B on WebQSP and have conducted careful inspection and analysis.
> >
> > After in-depth investigation, we found that this abnormal result is mainly caused by **format issues rather than model capability issues, and the core conclusions of the paper are not affected**. Specifically, Qwen2-1.5B failed to meet the expected output format requirements during the training process, resulting in its generated answers being marked as incorrect by the evaluation system due to non-standard formatting, even though the content is correct. For example:
> >
> >     "question": "During which NFL season did the team owned by Steve Biscotti win the super bowl?"
> >     "prediction": I can assist you with your question. 😊\n# Reasoning Path:\nSuper bowl -> freebase.type_profile.equivalent_topic -> Super Bowl -> sports.sports_championship.events -> Super Bowl XXXV\n# Answer:\nSuper Bowl XXXV
> >     "ground_truth_paths": "Super bowl -> freebase.type_profile.equivalent_topic -> Super Bowl -> sports.sports_championship.events -> Super Bowl XXXV"
> >
> > This phenomenon is relatively common in format-sensitive tasks.
> >
> > To ensure the rigor of the experiment, we adopted strictly consistent training and evaluation settings for Qwen2-1.5B as other models, including: the same data recipe, learning rate and training steps, format consistency, template verification and constrained decoding, etc. Based on these improvements, we re-conducted the experiment, and the **updated results have been reflected in Tab. 4** as follows:
> >
> > | Backbone | WebQSP |  | CWQ |  |
> > |---|---|---|---|---|
> > |  | Hit@1 | F1 | Hit@1 | F1 |
> > | Microsoft series |  |  |  |  |
> > | Phi-3-mini-4k-instruct | 83.1 | 65.7 | 69.5 | 50.2 |
> > | Mistralai series |  |  |  |  |
> > | Mistral-7B-Instruct-v0.3 | 95.2 | 61.8 | 77.2 | 48.3 |
> > | Internlm series |  |  |  |  |
> > | internlm2_5-7b-chat | 95.0 | 65.4 | 75.8 | 46.5 |
> > | internlm2-chat-7b | 95.6 | 65.3 | 75.6 | 51.7 |
> > | Qwen series |  |  |  |  |
> > | Qwen-2-0.5b | 81.6 | 60.4 | 75.2 | 35.9 |
> > | Qwen2-1.5b | 92.8 | 60.5 | 73.1 | 41.0 |
> > | Qwen-2-7b | 95.9 | 60.6 | 75.8 | 43.0 |
> > | Qwen2.5-14B-Instruct | 96.6 | 71.5 | 77.0 | 54.8 |
> > | Meta-llama series |  |  |  |  |
> > | LLaMA-2-7b | 94.0 | 58.2 | 82.8 | 41.6 |
> > | LLaMA-3.1-8b | 95.4 | 63.3 | 80.9 | 45.0 |
> > | Llama-2-13b-chat-hf | 96.0 | 75.2 | 83.1 | 64.1 |
> >
> > To ensure the reproducibility of the experiment, we have added corresponding configuration files, training logs, and evaluation scripts to the code repository, enabling the academic community to verify and reproduce our results.

---

> > ### Comment · Reviewer_EXWb · 2025-11-24
> >
> > Thank you for your response.
> >
> > I am still confused why you state "we conduct reasoning using two A800 80G GPUs" in line 898 assuming that it does not introduce much additional cost and memory.
> >
> > In addition, what is "video memory usage"?
> >
> > As for the revision, I hope that "the difference between logic drift and hallucination" can be discussed in the main paper. Try to make the tables on page 9 be laid out well without so many blank spaces.

---

> > > ### Author Response · Authors · 2025-11-24
> > > **Response to Official Comment by Reviewer EXWb**
> > >
> > > Thank you for your response and constructive suggestions. We address your concerns point by point below:
> > >
> > > **Regarding computational cost and memory usage**:
> > >
> > > To clarify why our method does not introduce much additional cost and memory, we conducted comprehensive experiments using two A800 80G GPUs on LLaMA3-8b with the following configurations: (1) direct inference on original LLaMA3-8b without Logits-to-Logic, (2) LLaMA3-8b with only the $\mathcal{Z}\_f$ module, (3) LLaMA3-8b with only the $\mathcal{Z}_s$ module, and (4) LLaMA3-8b with complete Logits-to-Logic (both $\mathcal{Z}_f$ and $\mathcal{Z}_s$).
> > >
> > > | Method | GPU Usage (GB) | Running Time (h) |  |
> > > |---|---|---|---|
> > > |  |  | CWQ | WebQSP |
> > > | LLaMA3-8b | 29.01 | 15.28 | 7.25 |
> > > | LLaMA3-8b w/ $\mathcal{Z}_f$ | 29.01 | 8.11 | 6.41 |
> > > | LLaMA3-8b w/ $\mathcal{Z}_s$ | 29.09 | 26.33 | 11.06 |
> > > | LLaMA3-8b w/ Logits-to-Logic | 32.89 | 18.63 | 8.48 |
> > >
> > > The experimental results demonstrate that: **For GPU memory usage**, Logits-to-Logic requires a total of 32GB (16GB per A800 GPU), adding only approximately 3GB compared to direct LLaMA3-8b inference. **For running time**, Logits-to-Logic adds only 3 hours (+21.9%) and 1 hour (+16.9%) on CWQ (3,531 samples) and WebQSP (1,628 samples) respectively. **These results confirm that Logits-to-Logic introduces minimal additional memory and computational overhead.**
> > >
> > > To further substantiate our computational efficiency claims, we conducted a comprehensive comparison of computational overhead **between Logits-to-Logic and advanced methods**. The results demonstrate that **our approach exhibits substantial advantages in terms of LLM API call frequency and computational expenses compared to existing methods**:
> > >
> > > | Type | Model | CWQ |  |  | WebQSP |  |  |
> > > |---|---|---|---|---|---|---|---|
> > > |  |  | # LLM Call | Total Token | Total Cost | # LLM Call | Total Token | Total Cost |
> > > | LLMs Reasoning | CoT | 1 | 409.7 | 0.00008 | 1 | 397.6 | 0.00008 |
> > > | Agentic Reasoning | ToG | 9.2 | 11468.5 | 0.0023 | 8.8 | 10189.4 | 0.0021 |
> > > |  | DoG | 5.7 | 37919.7 | 0.006 | 2.7 | 6114.5 | 0.001 |
> > > |  | GCR | 2 | 125.3 | - | 2 | 231.0 | - |
> > > |  | KG-CoT | - | - | - | 1 | - | 0.020 |
> > > |  | PoG | 13.3 | 8,156.2 | 0.0016 | 9.0 | 5,517.7 | 0.0011 |
> > > | Ours | Logits-to-Logic | 1 | 1270.8 | - | 1 | 716.9 | - |
> > >
> > > **Regarding the terminology clarification**:
> > >
> > > We apologize for the **confusion regarding "video memory usage".** We have corrected this terminology in the revised **Authors' Response Part 4 (Response to W4)**. The accurate statement should be:
> > >
> > >     Experiments show that on LLaMA3-8b, our method only requires an additional (~3GB) of GPU memory usage, and on CWQ and WebQSP, it increases the running time by an additional 3 hours and 1 hour respectively.
> > >
> > > **Regarding the requested revisions**:
> > >
> > > We have **updated the revised version of our paper** with the following modifications:
> > >
> > > (1) Added new discussing "Differences between Logic Drift and Hallucination" in the **main paper (Lines 53-56)**, with additional detailed discussion provided in **Appendix L**.
> > >
> > > (2) Improved the layout of Table 4 on page 9 to minimize blank spaces and enhance readability.
> > >
> > > **We genuinely hope that our responses and revisions have adequately addressed the concerns and suggestions raised in your initial review and follow-up comments. Please do not hesitate to let us know if there is anything further we could do to improve your impression and final rating of our work.**

---

> > > > ### Comment · Reviewer_EXWb · 2025-11-25
> > > >
> > > > No, please read my question carefully. I just wonder why you state "we conduct reasoning using two A800 80G GPUs" in line 898, assuming that it does not introduce much additional cost and memory.
> > > >
> > > > I noticed that the table showing your method only takes 32GB memory in your original response (where I have said it does not introduce much additional cost and memory). But why still need to run on **two** A800 GPUs with 80G memory?
> > > >
> > > > I think this is not a good response where you misunderstood the question and just repeatedly put the contents already exist in the response letters (response to W4), making the discussion very tedious.

---

> > > > > ### Author Response · Authors · 2025-11-25
> > > > >
> > > > > Thank you for pointing out the correct understanding. Logits-to-Logic requires only 32GB memory on LLaMA3-8B, meaning it can run on a single GPU with 32GB capacity. **We reported using 2 A800 GPUs because our server configuration is an A800 cluster**. Actually, the two GPUs were used only for higher throughput and faster experimentation, not as a method requirement. Smaller GPUs can be used for deployment.
> > > > >
> > > > > Thank you for your valuable feedback. If you have any other questions or would like to discuss our response further, please don't hesitate to reply. We would be very happy to engage in deeper discussions with you.

---

### Official Review · Reviewer_nv8o · 2025-10-28

**Soundness:** 3
**Presentation:** 3
**Contribution:** 3
**Rating:** 6
**Confidence:** 4

**Summary:**

The manuscript proposes a novel framework, Logits-to-Logic, to enhance teh logical reasoning capabilities of LLM when applied to KGs. The main problem tackled by this manuscript is that LLM struggle with Logic Drift, where their reasoning paths often do not align with the logical structure of the KG, leading to errors in structured knowledge reasoning.

The authors propose Logits-to-Logic, a framework that directly operate on the logits output by the LLM during their autoregressive generation. Logits-to-Logic consists logic compiling, logits trenghthening, logits filtering.

**Strengths:**

+ The manuscript addresses the issue of logic drift by directly intervening in the logits.
+ Extensive experiments show significant improvements.
+ The framework demonstrates significant computational efficiency

**Weaknesses:**

- Evaluation mainly focus on LLaMa and Qwen model. Other foundation models or larger models are not validated due to resource constraints. It would benefit analyzing how the framework scales with larger models.
- Although the class-agnostic loss helps prevent overfitting to specific classes, the overall framework may stil struggle with class imbalance or biased training samples
- The framework relies heavily on the predefined hyperparameters for the loss terms. While the paper show an empirical study on these values, further dynamic or adaptive tuning methodology may benefit the flexibility of the framework.

**Questions:**

Please refer weakness

---

> ### Author Response · Authors · 2025-11-20
> **Authors' Response Part 1**
>
> Dear Reviewer nv8o,
>
> Thank you for your thoughtful and insightful feedback. We sincerely appreciate your recognition of our **novel approach to address logic drift through direct logits intervention**, the **significant improvements demonstrated in our extensive experiments**, and the **computational efficiency** of our Logits-to-Logic framework. We have carefully addressed the issues you raised regarding the weaknesses (W) and questions (Q) of our paper. We hope our response (R) provides clarity and enhances your overall impression of our work.
>
> **Response to W1**:
>
>     W1: Limited to LLaMa and Qwen models due to resource constraints, the evaluation lacks broader foundation model validation and scalability analysis for larger models.
>
> Thank you for your constructive suggestion. We conducted experiments on LLM backbones from other series (in addition to the LLaMA and Qwen series, we **added models from Microsoft, Mistralai, and Internlm series**) and **larger parameter models (Qwen2.5-14B-Instruct and Llama-2-13b-chat-hf)**. The experimental results are as follows:
>
> | Backbone | WebQSP |  | CWQ |  |
> |---|---|---|---|---|
> |  | Hit@1 | F1 | Hit@1 | F1 |
> | Microsoft series |  |  |  |  |
> | Phi-3-mini-4k-instruct | 83.1 | 65.7 | 69.5 | 50.2 |
> | Mistralai series |  |  |  |  |
> | Mistral-7B-Instruct-v0.3 | 95.2 | 61.8 | 77.2 | 48.3 |
> | Internlm series |  |  |  |  |
> | internlm2_5-7b-chat | 95.0 | 65.4 | 75.8 | 46.5 |
> | internlm2-chat-7b | 95.6 | 65.3 | 75.6 | 51.7 |
> | Qwen series |  |  |  |  |
> | Qwen-2-0.5b | 81.6 | 60.4 | 75.2 | 35.9 |
> | Qwen2-1.5b | 92.8 | 60.5 | 73.1 | 41.0 |
> | Qwen-2-7b | 95.9 | 60.6 | 75.8 | 43.0 |
> | Qwen2.5-14B-Instruct | 96.6 | 71.5 | 77.0 | 54.8 |
> | Meta-llama series |  |  |  |  |
> | LLaMA-2-7b | 94.0 | 58.2 | 82.8 | 41.6 |
> | LLaMA-3.1-8b | 95.4 | 63.3 | 80.9 | 45.0 |
> | Llama-2-13b-chat-hf | 96.0 | 75.2 | 83.1 | 64.1 |
>
> The experimental results show that Llama-2-13b-chat-hf and Qwen2.5-14B-Instruct achieved better performance. Meanwhile, we have supplemented this part of the experimental results to **Tab. 4** in the main text.
>
> **Response to W2 (R2)**:
>
>     W2: Despite the class-agnostic loss reducing class-specific overfitting, the framework may still struggle with class imbalance or biased samples.
>
> Thank you for your attention to the training part of our method. Overall, Logits-to-Logic focuses on correcting logits distribution during LLMs decoding to achieve precise reasoning, which is a **training-free** process. Prior to this, we hope that LLMs output textual path formats that meet expectations (each entity in KG paths is connected using ->, see Fig.7), as shown below:
>
>     Help Me Make It Thru the Night -> common.topic.notable_types -> Composition -> type.type.properties -> Lyricist
>     Help Me Make It Thru the Night -> music.composition.composer -> Joe Walsh -> music.guitarist.guitars_played -> Fender Stratocaster
>     Help Me Make It Thru the Night -> common.topic.notable_types -> Composition -> type.type.properties -> Composer ...
>
> This standardized path output format makes it more convenient for us to correct the logits distribution of each token in each path.
>
> Therefore, we use 1/10 of the data from CWQ and WebQSP training sets to perform SFT training on LLM (as mentioned in Implementation Details, Line 326-327), but the purpose of this training is only to make LLMs learn the expected output format (while strictly ensuring data isolation and not bringing any prior knowledge of the test set to the LLM), which is not the focus of our method. Therefore, we need to specifically clarify:
>
> 1.Our SFT before decoding is only aimed at making LLM learn the expected output format, rather than learning category discrimination or path content priors. Specifically, this SFT only optimizes the general autoregressive language modeling loss, **allowing the model to mimic the correct output format**; **this process does not involve category label sampling or weighting, nor does it bias toward certain types of relations/types, so it will not introduce or exacerbate class imbalance problems**
>
> 2.The SFT part is not the focus of our method. Our research focus is on how to essentially alleviate the Logic Drift problem from the perspective of LLM output. **Class imbalance/biased samples are not the problems we study, nor are they considered in our research scenario**

---

> ### Author Response · Authors · 2025-11-20
> **Authors' Response Part 2**
>
> **Response to W3**:
>
>     W3: The framework relies heavily on predefined hyperparameters for loss terms. Despite empirical validation, dynamic or adaptive tuning methods could improve framework flexibility.
>
> Our Logits-to-Logic framework **only involves one hyperparameter**, namely the strength value $\omega$ in stage 2——Logits Strengthening, which controls the enhancement strength of the logits values of correct tokens during LLM decoding process, and this parameter is **unrelated to training** (as explained in R2 above, our SFT does not involve any parameters). Meanwhile, we have already conducted experiments on the impact of hyperparameter $\omega$ in Fig.4 Right, and the experiments confirmed that our chosen $\omega=2.0$ is the optimal value. We also report the impact of $\omega$ value selection on performance again below:
>
> | Strength Value | CWQ |  | WebQSP |  |
> |---|---|---|---|---|
> |  | Hit@1 | F1 | Hit@1 | F1 |
> | 10.0 | 70.47 | 27.80 | 91.35 | 45.17 |
> | 5.0 | 74.55 | 42.91 | 94.89 | 58.92 |
> | 3.0 | 80.32 | 42.69 | 96.08 | 62.61 |
> | 2.0 | 80.87 | 44.95 | 95.40 | 63.29 |
> | 1.5 | 73.60 | 42.69 | 96.13 | 65.46 |
> | 1.0 | 73.96 | 45.27 | 93.93 | 64.83 |
> | -1.0 | 68.43 | 36.89 | 90.90 | 60.92 |
>
> More detailed analysis about $\omega$ has been explained in Line 367-380 of the paper.
>
> In summary, our Logits-to-Logic only involves one hyperparameter $\omega$, and **does not heavily rely on multiple predefined hyperparameters, nor does it require complex methods for automated parameter adjustment**. Meanwhile, our method with $\omega=2.0$ **can adapt to different structured knowledge reasoning tasks** (single hop, multi-hop, slot filling) and **different scales of knowledge graphs** (Freebase and large-scale Wikidata) without any adjustment, achieving advanced performance in all cases. The transfer experiments in Tab. 3 have confirmed this point, which also demonstrates that our method possesses **excellent flexibility, transferability and robustness.**

---

### Official Review · Reviewer_bu9C · 2025-10-31

**Soundness:** 2
**Presentation:** 1
**Contribution:** 2
**Rating:** 4
**Confidence:** 2

**Summary:**

This paper proposes a knowledge graph question answering method based on large language models (LLMs). When using LLMs to answer knowledge graph (KG) questions, they sometimes generate paths that do not exist in the KG or produce incorrect paths, leading to wrong answers. To address this, the authors propose aligning the logits of the LLM output with the logical structure of the knowledge graph to ensure the generated answers are faithful. Experimental results show that the proposed method outperforms existing state-of-the-art approaches.

However, the paper is somewhat difficult to understand, particularly because it lacks an introductory section or preliminary knowledge, making it challenging for readers to assess its contributions. Moreover, the contribution appears somewhat incremental.

I will try to review the paper again during the rebuttal period, so I hope the authors can provide sufficient information at that time.

**Strengths:**

The idea of aligning the logits of LLM outputs with knowledge graph logic to ensure that LLM outputs follow KG information is interesting. However, the idea is somewhat difficult to understand. LLM outputs are probability distributions over tokens, while in a KG, an entity name or relation may consist of multiple words. It is unclear how this mapping is performed.

The experimental results are promising, but in Table 1, only Hit@1 is reported; F1 scores are not provided. It would be better to include the F1 scores for all baseline methods for a more comprehensive comparison.

**Weaknesses:**

The paper lacks an introduction to preliminary knowledge, which makes it difficult to understand and to assess the true contributions. Moreover, the contribution seems somewhat incremental.

No example is provided in the paper. It would be helpful to include a complete example that illustrates the entire process, from beginning to end, to facilitate understanding.

**Questions:**

no

---

> ### Author Response · Authors · 2025-11-20
> **Authors' Response Part 1**
>
> Dear Reviewer bu9C,
>
> Thank you for your thoughtful and insightful feedback. We sincerely appreciate your recognition of **the interesting nature of our approach to align LLM logits with knowledge graph logic to ensure faithful answer generation**, as well as your acknowledgment of our **promising experimental results**. We have carefully addressed the issues you raised regarding the weaknesses (W) and questions (Q) of our paper. We hope our response (R) provides clarity and enhances your overall impression of our work.
>
> **Response to W1**:
>
>     W1: The paper lacks background introduction, hindering comprehension and contribution assessment. Moreover, the contribution seems somewhat incremental.
>
> Thank you for your valuable feedback. We address each concern below:
>
> **Background and Problem Definition**: Knowledge Graph Question Answering (KGQA) involves answering natural language questions using structured factual information stored in knowledge graphs (KGs). A knowledge graph $G$ is organized as a collection of triplets: $G = \{(e_s, r, e_o) \in E \times R \times E\}$, where $E$ is the entity set and $R$ is the relation set. Multiple consecutive triplets with matching head-tail entities form reasoning paths in $G$, defined as:
> $s = e_1 \rightarrow r_1 \rightarrow e_2 \rightarrow r_2 \rightarrow e_3 \rightarrow ... \rightarrow r_{l-1} \rightarrow e_l, \forall e_i \in E, r_i \in R$.
> These reasoning paths provide precise and explainable answers by connecting question entities to answer entities through structured relationships in the KG.
>
> **Logic Drift Problem and Existing Limitations**: Large Language Models (LLMs) excel at natural language reasoning but struggle with structured knowledge due to representational differences between unstructured text and structured KGs. This leads to Logic Drift: LLMs generate reasoning paths that either (1) don't exist in the KG, or (2) are semantically irrelevant to the question intent logic.
>
> Existing methods address this by designing complex agent-based workflows embedded in prompts (e.g., ToG, DoG, KG-CoT). However, these approaches only provide input-level guidance and fail to fundamentally resolve Logic Drift in LLM outputs, as evidenced by persistent high Logic Drift ratios across current methods (As shown in Fig.1 in the paper).
>
> **Our Key Innovation**: Despite these efforts, existing methods still suffer from severe Logic Drift because they only provide input-level guidance while lacking flexibility and transferability. We identify that Logic Drift stems from inconsistency between LLMs' output logits distributions and the logical distributions of the structured KG and question. By analyzing LLMs' last-layer logits, we observe that tokens corresponding to incorrect reasoning paths have high logits values, while correct tokens have low values.
>
> Unlike previous methods that design complex workflows or prompt engineering, we address Logic Drift from the output perspective, providing deeper insights into logic-consistent reasoning. This enables us to map question and KG logic into the logits probability space. Our Logits-to-Logic framework can flexibly transfer to different tasks and KGs without modifications, representing the core contribution of our work.
>
> **Our Solution - Logits-to-Logic Framework**: Unlike existing input-level approaches, we directly target LLMs' output process. Our framework operates on the last-layer logits through two core modules:
>
> Logits Strengthening: Enhances logits values of tokens that align with question semantic logic using differentiation between original and masked prompts
>
> Logits Filtering: Constrains logits of illegal tokens that don't belong to valid KG paths using Non-deterministic Finite Automaton (NFA) transition functions
>
> **Below we continue with our response to W1.**

---

> > ### Author Response · Authors · 2025-11-20
> > **Authors' Response Part 2**
> >
> > **Response to W1 (continued)**:
> >
> >     W1: The paper lacks background introduction, hindering comprehension and contribution assessment. Moreover, the contribution seems somewhat incremental.
> >
> > **Complete Example**:
> >
> > Consider the question: "What kind of guitar was used by the lyricist for Help Me Make It Thru the Night?"
> >
> > 1. Logic Compiling - Building NFA:
> >
> > (1) Extract topic entity: "Help Me Make It Thru the Night"
> >
> > (2) Compile all valid KG paths starting from this entity into NFA states:
> >
> > Path 1: Help Me Make It Thru the Night -> music.composition.composer -> Joe Walsh -> music.guitarist.guitars_played -> Fender Stratocaster
> >
> > Path 2: Help Me Make It Thru the Night -> common.topic.notable_types -> Composition -> type.type.properties -> Composer
> >
> > Path 3: Help Me Make It Thru the Night -> film.composition.language -> English Language -> human.language.
> > countries_spoken_in -> Vatican City
> >
> > Path 4: Help Me Make It Thru the Night -> art.painting.artist -> ...
> >
> > (3) Create NFA states: Each partial path becomes a state (e.g., $State_1$: "Help Me Make It Thru the Night", $State_2$: "Help Me Make It Thru the Night → common.topic.notable", $State_3$: "Help Me Make It Thru the Night → common.topic.notable_types")
> >
> > (4) Define transition function $\delta$: For each state, $\delta$ specifies valid next tokens (e.g., from $State_2$, only "_types" is valid)
> >
> > (5) Score paths with sentence-transformer: Rank paths by semantic similarity to question intent, identifying the path as high-scoring (Path 1) and other paths as low-scoring (Path 2~4)
> >
> > 2. Logits Strengthening using NFA: When generating token after "Help Me Make It Thru the Night -> ":
> >
> > Original prompt: Contains high-scoring NFA paths (Path 1)
> >
> > Masked prompt: Replaces high-scoring paths (Path 1) with MASK tokens, leaving only low-scoring noise paths (Path 2~4)
> >
> > NFA guidance: High-scoring paths indicate "music" should be strengthened
> >
> > Operation: Calculate the difference between original and masked output probabilities, multiply by coefficient $\omega$, then add back to the masked probabilities. This amplifies correct token "music" logits while reducing irrelevant tokens like "film" or "art".
> >
> > 3. Logits Filtering using NFA: At each generation step:
> >
> > Current state tracking: Monitor current position in NFA (e.g., currently at $State_2$)
> >
> > Valid token identification: Use $\delta$ to identify legal next tokens from current state
> >
> > Filtering operation: Set logits of illegal tokens to $-\infty$ values
> >
> > Example: From $State_2$, if LLM tries to generate "award" (not in $\delta$ transitions), set its logits to $-\infty$, forcing selection among valid tokens like "_types"
> >
> > About Token-level Mapping: Since entities/relations may contain multiple words, we handle this by:
> >
> > Tokenization: Split multi-word entities (e.g., "Help Me Make It Thru the Night" → ["Help", "Me", "Make", "It", "Thru", "the", "Night"])
> >
> > State tracking: NFA tracks partial token sequences, allowing transitions on individual tokens
> >
> > Completion validation: Only complete entity/relation names in NFA are considered valid states
> >
> > Result: The framework **compiles question intent and KG logic into NFA to guide LLM**, generating the logic-consistent reasoning path (Path 1: Help Me Make It Thru the Night -> music.composition.composer -> Joe Walsh -> music.guitarist.guitars_played -> Fender Stratocaster) to identify "Fender Stratocaster" as the answer while preventing Logic Drift at each token generation step.
> >
> > If you have any further questions, please feel free to ask :).
> >
> > **Response to W2**:
> >
> >     W2: In Table 1, only Hit@1 is reported; F1 scores are not provided.
> >
> > Thank you for your suggestion, which is very constructive for comprehensively demonstrating the performance comparison of our method. Therefore, we have **supplemented the F1 score comparison between Logits-to-Logic and existing methods**. Since some agentic reasoning methods did not report F1 metrics in their papers, we present below the comparison with methods that reported F1 metrics (all data sourced from their respective papers):
> >
> > | Method | WebQSP | CWQ |
> > |:---:|:---:|:---:|
> > | LLM Reasoning |  |  |
> > | Qwen2-0.5b | 17.2 | 11.0 |
> > | Qwen2-1.5b | 28.0 | 15.7 |
> > | Qwen2-7b | 35.5 | 21.6 |
> > | LLaMA2-7b | 36.5 | 21.4 |
> > | LLaMA3.1-8b | 34.8 | 22.4 |
> > | GPT-4o-mini | 40.5 | 40.5 |
> > | ChatGPT | 43.5 | 30.2 |
> > | ChatGPT+Few-shot | 38.1 | 28.0 |
> > | ChatGPT+CoT | 38.5 | 31.0 |
> > | Agentic Reasoning |  |  |
> > | KD-CoT | 52.5 | - |
> > | RoG | 70.8 | 56.2 |
> > | GCR | 74.1 | 61.7 |
> > | Logits-to-Logic  w/ LLaMA3-8b | 63.3 | 45.0 |
> > | Logits-to-Logic w/ LLaMA3-13b | 75.2 | 64.1 |
> >
> > Based on the comprehensive experimental results in Tab. 1, our method demonstrates **comprehensive advantages in both Hit@1 and F1**. Meanwhile, we have **supplemented these experimental results to Appendix C** in the main text.

---

> ### Comment · Reviewer_bu9C · 2025-11-20
> **comments**
>
> Thank you for the authors’ feedback. I now have a better understanding of the paper, but I still have several questions:
>
> 1. Since LLMs generate tokens one by one, how can you ensure that “from $State_2$, only _types is valid”? the word "_types" itself contains multiple tokens, and I’m not very clear about how this step works. What if LLMs generate other tokens instead of  "_types"?
>
> 2. What is the main difference between the approach in this paper and the idea in “Graph-constrained Reasoning: Faithful Reasoning on Knowledge Graphs with Large Language Models”? It seems like they are very similar.
>
> 3. Regarding the experimental results: the F1 scores do not seem very high, especially for Logits-to-Logic w/ LLaMA3-8b, which does not outperform many baselines. While Logits-to-Logic w/ LLaMA3-13b can outperform some baselines, but those baselines use LLaMA3-8b instead of LLaMA3-13b.

---

> > ### Author Response · Authors · 2025-11-21
> > **Response to comments by Reviewer bu9C Part 1**
> >
> > Thank you for your thoughtful questions, which help clarify important technical details and positioning of our work. We address each concern below:
> >
> > ## 1.Multi-token Entity Handling in NFA
> >
> > Thank you for your excellent technical question about handling token like "_types".
> >
> > Clarification on the Specific Example: In our example path "Help Me Make It Thru the Night -> common.topic.notable_types -> Composition -> type.type.properties -> Composer", the tokenization actually produces (e.g., using Qwen as tokenizer):
> >
> >     'Help' | ' Me' | ' Make' | ' It' | ' Th' | 'ru' | ' the' | ' Night' | ' ->' | ' common' | '.topic' | '.not' | 'able' | '_types' | ' ->' | ' Composition' | ' ->' | ' type' | '.type' | '.properties' | ' ->' | ' Composer'
> >
> > As shown, "_types" is tokenized as a **single token** in this path, which is why our filtering mechanism can directly constrain it. From $State_2$, if the LLM tries to generate "award" (not in $\delta$ transitions), we set its logits to $-\infty$, forcing selection among valid tokens like "_types".
> >
> > General Multi-token Handling: For entities that do span multiple tokens (like "Help Me Make It Thru the Night"), our NFA handles this through:
> >
> > - Tokenization: Multi-word entities are split into individual tokens ["Help", "Me", "Make", "It", "Thru", "the", "Night"]
> > - State tracking: NFA tracks partial token sequences, allowing transitions on individual tokens
> > - State validation: Only complete entity/relation names in NFA are considered valid final states
> >
> > This design ensures token-by-token enforcement while correctly handling both single-token and multi-token entities in KG paths.
> >
> > If you have any further questions about this mechanism, please feel free to discuss! :)
> >
> >
> > ## 2.Key Differences from GCR (Graph-constrained Reasoning)
> >
> > Thank you for this excellent question that helps clarify our unique contributions. While both methods aim to improve LLM reasoning on KGs, they represent fundamentally different paradigms.
> >
> > Core Paradigm Differences:
> >
> > - GCR: Employs a dual-agent approach with "KG expert + LLM" using KG Trie to constrain the expert to real paths, then LLM reasoning on verified paths. However, the KG expert lacks contextual information when proposing candidates, potentially generating paths inconsistent with question intent. **Essentially, it follows the input-level paradigm similar to KG-CoT, using KG's real paths as enhanced context to guide LLM reasoning**.
> > - Our Logits-to-Logic: We identify that Logic Drift stems from inconsistency between LLMs' output logits distributions and the logical distributions of structured KG and question intent. By analyzing LLMs' last-layer logits, we observe that tokens corresponding to incorrect reasoning paths have high logits values, while correct tokens have low values. **Therefore, we address Logic Drift from the output perspective, providing deeper insights into logic-consistent reasoning by mapping question and KG logic into the logits probability space**.
> >
> > Experimental Evidence of Superior Logic Drift Reduction (detailed in Fig. 1 and Fig. 4 Right in the paper): We analyzed Logic Drift patterns across **input-level methods including GCR** (C1 and C2 correspond to KG-Inconsistent and Question Intent Inconsistent Logic Drift, respectively):
> >
> > | Method | C1 | C2 | Total Logic Drift |
> > |---|---|---|---|
> > | Ours | 4 | 10 | 14 |
> > | GCR | 0 | 27 | 27 |
> > | ToG | 31 | 30 | 61 |
> > | KG-CoT | 4 | 30 | 34 |
> >
> > **Our method achieves significantly lower logic drift (14 vs GCR's 27), demonstrating the effectiveness of output-level alignment over input-level constraints.**
> >
> > Comprehensive Logic Alignment: **Unlike GCR's focus solely on KG path hallucinations, we take a broader perspective addressing both question intent and KG intrinsic logic through our dual modules (Logits Strengthening $\mathcal{Z}\_s$ and Logits Filtering $\mathcal{Z}_f$)**:
> >
> > | Module | WebQSP Hit@1 Gain | CWQ Hit@1 Gain | Function |
> > |---|---|---|---|
> > | $\mathcal{Z}_s$  | +63.0% | +23.9% | Enhances question intent alignment |
> > | $\mathcal{Z}_f$  | +49.6% | +33.1% | Enforces KG structural constraints |
> >
> > Both modules contribute substantially, with $\mathcal{Z}_s$ addressing question logic and $\mathcal{Z}_f$ handling KG logic, providing comprehensive logic-consistent reasoning.
> >
> > **Deeper Technical Insights**: Our logits distribution visualization (shown in Fig. 5 in the paper) reveals the fundamental importance of aligning LLM output distributions for logic-consistent reasoning, **offering novel insights that input-level methods (e.g. GCR, KG-CoT) cannot provide.**
> >
> > In summary, while GCR constrains inputs through "KG expert + LLM" workflow design, Logits-to-Logic fundamentally aligns output probability distributions with structured logic, representing a paradigm shift from input-level guidance to output-level control for more effective Logic Drift mitigation.

---

> ### Author Response · Authors · 2025-11-21
> **Response to comments by Reviewer bu9C Part 2**
>
> ## 3.F1 Score Analysis and Fair Comparison
>
> Thank you for raising this important point about experimental comparison fairness. We need to clarify a crucial detail regarding GCR's experimental setup.
>
> Critical Clarification on GCR's Configuration: GCR uses **LLaMA3.1-8B + GPT-4o-mini** in their experiments (as detailed in their original paper), meaning their F1 scores are achieved through a hybrid system leveraging OpenAI's advanced closed-source model. In contrast, our F1 scores are obtained **using only LLaMA3.1-8B or LLaMA3-13B without any external model assistance**.
>
> Fair Performance Analysis:
> - Single Model vs. Hybrid System: Our LLaMA3.1-8B naturally cannot match GCR's LLaMA3.1-8B + GPT-4o-mini F1 scores, as they utilize a more advanced closed-source model
> - Superior Hit@1 Performance: **Even with single LLaMA3.1-8B, we achieve state-of-the-art Hit@1 scores that surpass GCR's hybrid system**
> - Competitive F1 with Fair Comparison: **Our LLaMA3-13B achieves superior F1 scores compared to baseline methods using comparable model sizes**
>
> Key Advantages Demonstrated:
> - **Consistent Cross-Scale Effectiveness**: Our method shows improvements across model sizes (0.5B to 14B), achieving advanced performance even on smaller models
> - **Resource Efficiency (detailed in Tab. 8 in paper)**: We achieve competitive/superior performance using single models versus baselines requiring hybrid systems or multiple agents
> - **Superior Transferability (detailed in Tab. 3 in paper)**: Most importantly, our method transfers seamlessly to unseen large-scale Wikidata KGs and different reasoning tasks without any modification or training, maintaining advanced performance with LLaMA3.1-8B:
>
> | Method | Muti-hop | Slot Filling |  |
> |---|---|---|---|
> |  | QALD10-en (Wikidata) | T-REx (Wikidata) | Zero-shot RE (Wikidata) |
> | LLMs Reasoning |  |  |  |
> | IO prompt w/ChatGPT | 42.0 | 33.6 | 27.7 |
> | CoT w/ChatGPT | 42.9 | 32.0 | 28.8 |
> | SC w/ChatGPT | 45.3 | 41.8 | 45.4 |
> | Agentic Reasoning |  |  |  |
> | ToG-R ToG w/ ChatGPT | 48.6 | 75.3 | 86.5 |
> | ToG w/ ChatGPT | 50.2 | 76.8 | 88.0 |
> | ToG-R w/ GPT4 | 54.7 | 75.5 | 86.9 |
> | ToG w/ GPT4 | 53.8 | 77.1 | 88.3 |
> | Ours w/ LLaMA3.1-8B | **59.2** | **87.6** | **91.4**  |

---

> > ### Comment · Reviewer_bu9C · 2025-11-24
> >
> > Thank you for the authors’ detailed feedback.
> >
> > 1. If I understand correctly, the proposed method constructs an NFA over the entire knowledge graph and then performs output-logit adaptation at every step of the LLM’s generation process. I therefore have some concerns regarding the time complexity and computational overhead of this approach.
> >
> > 3. Thank you for the response. However, when referring to baseline methods, I was not referring specifically to GCR. I meant the many other baselines published in the past two years. Several of these methods appear to outperform the proposed approach when using LLaMA3-8B or even just LLaMA-2-7b.
> >
> > In addition, I believe the paper requires further polishing. Many important details are not clearly explained, and readers must invest considerable effort to fully understand the method.

---

> > > ### Author Response · Authors · 2025-11-26
> > >
> > > Thank you for your feedback and constructive suggestions. We address each of your concerns systematically below.
> > >
> > >     Concerns regarding the time complexity and computational overhead of this approach
> > >
> > > We appreciate your concern about computational efficiency. We have conducted **comprehensive analyses of both theoretical complexity (Appendices I & J) and practical overhead through extensive experiments**.
> > >
> > > Our analysis encompasses two primary categories of computational overhead:
> > >
> > > (1) NFA construction overhead
> > >
> > > (2) Logits-to-Logic operation overhead for logits manipulation
> > >
> > > **NFA Construction Overhead Analysis**: We have **supplemented additional experiments on the complexity of NFA construction** to illustrate its actual overhead. Our hardware configuration is:
> > >
> > > | Machine Info. | Value |
> > > |---|---|
> > > | OS | Ubuntu 20.04.1 LTS |
> > > | CPU | Intel Xeon Gold 6326, 64 cores, 40 threads, 2.90GHz |
> > > | Memory | 629GB DDR4, 2666MHz |
> > >
> > > On two widely used KGQA datasets, Lagre scale Freebase—CWQ and WebQSP, we report the NFA construction complexity:
> > >
> > > Basic information about the WebQSP and CWQ datasets:
> > >
> > > | Value | CWQ |  |  | WebQSP |  |  |
> > > |---|---|---|---|---|---|---|
> > > |  | min | max | avg | min | max | avg |
> > > | # relations per entity | 0 | 259 | 43.6 | 2 | 258 | 76.9 |
> > > | # 2-hop paths per question | 9 | 11011 | 2025.6 | 10 | 11713 | 2024.6 |
> > > | token length per 2-hop path | 13 | 524 | 29.7 | 13 | 524 | 20.9 |
> > >
> > > NFA construction overhead:
> > >
> > > | #Thread | CWQ(#samples=27639) |  |  | WebQSP (#samples=2826) |  |  |
> > > |---|---|---|---|---|---|---|
> > > |  | Avg. Running Time per sample (s) | Peak Memory Usage (GB) | Parallel Efficiency (%) | Avg. Running Time per sample (s) | Peak Memory Usage (GB) | Parallel Efficiency (%) |
> > > | 1 | 0.046 | 103.7 | 127 | 0.052 | 12.0 | 133 |
> > > | 2 | 0.042 | 103.7 | 152 | 0.049 | 12.0 | 165 |
> > > | 4 | 0.039 | 103.7 | 159 | 0.045 | 12.0 | 175 |
> > > | 8 | 0.038 | 103.7 | 166 | 0.043 | 12.0 | 178 |
> > > | 16 | 0.038 | 103.7 | 166 | 0.042 | 12.0 | 189 |
> > > | 32 | 0.036 | 103.8 | 168 | 0.039 | 12.0 | 192 |
> > >
> > > The experiments demonstrate the **feasibility of constructing NFAs on large-scale KGs without requiring high time and computational overhead**.
> > >
> > > **Logits-to-Logic Operation Overhead**: Logits-to-Logic mainly modifies the logits distribution during the decoding phase of LLM inference, so it does not increase any parameters of the model. It only requires an extra forward pass on the Mask Prompt to obtain the logits distribution of the Mask Prompt, which **we illustrate through the additional supplementary experiments below**.
> > >
> > > | Method | GPU Usage (GB) | Running Time (h) |  |
> > > |---|---|---|---|
> > > |  |  | CWQ | WebQSP |
> > > | LLaMA3-8b | 29.01 | 15.28 | 7.25 |
> > > | LLaMA3-8b w/ $\mathcal{Z}_f$ | 29.01 | 8.11 | 6.41 |
> > > | LLaMA3-8b w/ $\mathcal{Z}_s$ | 29.09 | 26.33 | 11.06 |
> > > | LLaMA3-8b w/ Logits-to-Logic | 32.89 | 18.63 | 8.48 |
> > >
> > > Experiments show that on LLaMA3-8b, our method only requires an additional (~3GB) of GPU memory usage, and on CWQ and WebQSP, it increases the running time by an additional 3 hours and 1 hour respectively.
> > >
> > > We conduct a comprehensive comparison of computational overhead between Logits-to-Logic and advanced methods. The results demonstrate that our approach exhibits **substantial advantages in terms of LLM API call frequency and computational expenses** compared to existing methods.
> > >
> > > | Type | Model | CWQ |  |  | WebQSP |  |  |
> > > |---|---|---|---|---|---|---|---|
> > > |  |  | # LLM Call | Total Token | Total Cost | # LLM Call | Total Token | Total Cost |
> > > | LLMs Reasoning | CoT | 1 | 409.7 | 0.00008 | 1 | 397.6 | 0.00008 |
> > > | Agentic Reasoning | ToG | 9.2 | 11468.5 | 0.0023 | 8.8 | 10189.4 | 0.0021 |
> > > |  | DoG | 5.7 | 37919.7 | 0.006 | 2.7 | 6114.5 | 0.001 |
> > > |  | GCR | 2 | 125.3 | - | 2 | 231.0 | - |
> > > |  | KG-CoT | - | - | - | 1 | - | 0.020 |
> > > |  | PoG | 13.3 | 8,156.2 | 0.0016 | 9.0 | 5,517.7 | 0.0011 |
> > > | Ours | Logits-to-Logic | 1 | 1270.8 | - | 1 | 716.9 | - |

---

> ### Author Response · Authors · 2025-11-26
>
> Regarding Baseline Comparisons with Recent Methods
>
>     However, I was referring to other recent baselines from the past two years, not specifically GCR. Several of these methods outperform the proposed approach using LLaMA3-8B or LLaMA-2-7B.
>
> Thank you for clarifying your concern about recent baselines. We have conducted an exhaustive literature review and compiled comprehensive comparisons with methods published in the past two years using LLaMA3-8B/LLaMA2-7B models.
>
> | Method | WebQSP |  | CWQ |  |
> |:---:|:---:|:---:|:---:|:---:|
> |  | F1 | Hit@1 | F1 | Hit@1 |
> | *Closed-source Business Model* |  |  |  |  |
> | ChatGPT | 43.5 | 59.3 | 30.2 | 34.7 |
> | ChatGPT+Few-shot | 38.1 | 68.5 | 28.0 | 38.5 |
> | ChatGPT+CoT | 38.5 | 73.5 | 31.0 | 47.5 |
> | *Hybrid System (Large-Small Model Collaboration)* |  |  |  |  |
> | GCR w/ LLaMA3-8b+ChatGPT [1] | 73.2 | 92.6 | 60.9 | 72.7 |
> | GCR w/ LLaMA3-8b+GPT4o-mini [1] | 74.1 | 92.2 | 61.7 | 75.8 |
> | GNN-RAG w/ LLaMA2-7b+GNN [2] | 71.3 | 80.6 | 59.4 | 61.7 |
> | GSR w/ T5-3b + LLaMA2-7b [3] | 58.9 | - | 27.6 | - |
> | *7-8b Model* |  |  |  |  |
> | Qwen2-7b | 35.5 | 50.8 | 21.6 | 25.3 |
> | LLaMA2-7b | 36.5 | 56.4 | 21.4 | 28.4 |
> | LLaMA3.1-8b | 34.8 | 55.5 | 22.4 | 28.1 |
> | Interactive-KBQA w/ Mistral-7B [4] | 43.5 | 45.0 | 39.9 | 44.0 |
> | KD-CoT w/ LLaMA2-7b [5] | 52.5 | 68.6 | - | 55.7 |
> | RoG w/ LLaMA2-Chat-7B [6] | **70.8** | 85.7 | **56.2** | 62.6 |
> | SubgraphRAG w/ Llama3.1-8B [7] | 67.9 | 81.2 | 43.0 | 47.5 |
> | Logits-to-Logic  w/ LLaMA3-8b | 63.3 | **95.4** | 45.0 | **80.9** |
> | *13b Model* |  |  |  |  |
> | Interactive-KBQA w/ LLaMA2-13b [4] | 54.8 | 56.2 | 42.5 | 45.6 |
> | RoG w/ LLaMA3-13b [6] | 73.6 | 89.1 | 63.2 | 68.3 |
> | Logits-to-Logic w/ LLaMA3-13b | **75.2** | **96.0** | **64.1** | **83.1** |
>
> The results show that based on 7-8B models, our method achieves state-of-the-art Hit@1 performance (95.4% on WebQSP, 80.9% on CWQ), outperforming Interactive-KBQA w/ Mistral-7B by 50.4% and 36.9% respectively; surpassing KD-CoT by 26.8% and 25.2% respectively; exceeding RoG by 9.7% and 18.3% respectively; and outperforming SubgraphRAG by 14.2% and 33.4% respectively. In terms of F1 scores, our method lags behind RoG by 7.5% and 11.2% respectively, while still maintaining superiority over other methods. To provide a more comprehensive comparison of F1 scores, we conducted experiments on 13B models. For F1 scores, we outperform RoG by 1.6% and 0.9% respectively, while for Hit@1 scores, we surpass RoG by 6.9% and 14.8% respectively. Our method surpasses RoG w/ LLaMA3-13B on both F1 and Hit@1 metrics. This demonstrates that **our approach exhibits more pronounced improvements on larger models** (0.5B-13B range).
>
> The results indicate that our method scales effectively with model size, achieving comprehensive advanced performance on larger models.
>
>     [1] Linhao Luo, Zicheng Zhao, Gholamreza Haffari, Yuan-Fang Li, Chen Gong, Shirui Pan. Graph-constrained Reasoning: Faithful Reasoning on Knowledge Graphs with Large Language Models. ICML 2025
>     [2] Costas Mavromatis, George Karypis. GNN-RAG: Graph Neural Retrieval for Large Language Model Reasoning. ACL 2025.
>     [3] Wenyu Huang, Guancheng Zhou, Hongru Wang, Pavlos Vougiouklis, Mirella Lapata, Jeff Z. Pan. Less is More: Making Smaller Language Models Competent Subgraph Retrievers for Multi-hop KGQA. EMNLP 2024
>     [4] Guanming Xiong, Junwei Bao, Wen Zhao. Interactive-KBQA: Multi-Turn Interactions for Knowledge Base Question Answering with Large Language Models. ACL 2024.
>     [5] Keheng Wang, Feiyu Duan, Sirui Wang, Peiguang Li, Yunsen Xian, Chuantao Yin, Wenge Rong, Zhang Xiong. Knowledge-Driven CoT: Exploring Faithful Reasoning in LLMs for Knowledge-intensive Question Answering. ArXiv 2023.
>     [6] Linhao Luo, Yuan-Fang Li, Gholamreza Haffari, Shirui Pan. Reasoning on Graphs: Faithful and Interpretable Large Language Model Reasoning. ICLR 2024.
>     [7] Mufei Li, Siqi Miao, Pan Li. Simple Is Effective: The Roles of Graphs and Large Language Models in Knowledge-Graph-Based Retrieval-Augmented Generation. ICLR 2025.

---

> > ### Author Response · Authors · 2025-11-26
> >
> > Regarding Paper Clarity and Detail Explanation
> >
> >     The paper requires clearer explanation of important details to improve readability.
> >
> > We acknowledge the importance of clear presentation and have significantly enhanced the paper's readability in the revised version. **While space constraints prevent including all implementation details in the main text, we have provided comprehensive explanations in the appendices with clear references from the main text**.
> >
> > **Key Improvements Made**:
> >
> > 1. **Enhanced Motivation Discussion**: To facilitate readers' understanding of Logic Drift, we compare it with LLM hallucination in the main text (Lines 53-57) and provide comprehensive elaboration in Appendix L.
> >
> > 2. **Expanded Related Work**: We added elaborations on path-based and KG constrained generation KGQA methods in Appendix A.
> >
> > 3. **Detailed Method Exposition**:
> >
> > (a) From a theoretical derivation perspective, we supplemented explanations in Appendix F on the core approach of our method—how to align question intent with KG logical distributions through logits manipulation.
> >
> > (b) We detailed the operations of core modules and algorithmic flows in Appendices G and H.
> >
> > (c) Meanwhile, we provided detailed implementation details in Appendix D. We revised unclear expressions, figures, and typos in the method section of the main text.
> >
> > 4. **Comprehensive Experimental Analysis**:
> >
> > (a) We improved the backbone experiments, and in Section 4.5 Tab. 4, we supplemented experiments on LLM backbones from other series (Microsoft, Mistralai, and Internlm series) and larger parameter models (Qwen2.5-14B-Instruct and Llama-2-13b-chat-hf).
> >
> > (b) We improved the F1 metric analysis of our method in Appendix C.
> >
> > (c) We supplemented theoretical and experimental analysis of NFA construction overhead in Appendices I and J, as well as analysis of overall memory usage and runtime overhead of the method.
> >
> > We believe **these enhancements provide readers with sufficient detail to understand both our core contributions and implementation specifics**.
> >
> > Thank you for your valuable feedback. If you have any other questions or would like to discuss our response further, please don't hesitate to reply. We would be very happy to engage in deeper discussions with you.

---

### Author Response · Authors · 2025-11-20
**Summary of Rebuttal Part 1**

We would like to express our sincere gratitude to all reviewers for their thoughtful and constructive feedback. We genuinely appreciate the time and effort invested in helping us refine our work. We are delighted that **our Logits-to-Logic approach is recognized for its interesting nature in aligning LLM logits with knowledge graph logic to ensure faithful answer generation** (`Reviewer bu9C`), **novel approach to address logic drift through direct logits intervention with significant improvements and computational efficiency** (`Reviewer nv8o`), being **the first KGQA method to consider output-level control through logits manipulation with promising results** (`Reviewer EXWb`), and **clear motivation with model-agnostic nature that can be plugged into any decoder without retraining** (`Reviewer nnUp`).

We believe Logits-to-Logic will make a highly impactful contribution to the KGQA field. Unlike existing methods that design complex agent-based workflows or prompt engineering to provide only input-level guidance, **our framework addresses the fundamental Logic Drift problem from the output perspective by directly manipulating LLMs' last-layer logits**. Our core innovation lies in identifying that Logic Drift stems from inconsistency between LLMs' output logits distributions and the logical distributions of structured KG and question intent. Through our two-stage approach—Logits Strengthening and Logits Filtering using NFA—**we provide deeper insights into logic-consistent reasoning while maintaining excellent flexibility and transferability** across different tasks and KGs without modifications.

**Our detailed responses address individual queries and suggestions from each reviewer in their initial concerns and follow-up questions**, covering the following major categories:

**1. Background and Technical Clarification**:
- We provided comprehensive background on KGQA with detailed examples and specifics of our method to help readers understand both our core contributions and implementation specifics, thereby addressing `Reviewer bu9C`'s concerns regarding important details
- Thoroughly discussed the relationship between Logic Drift and hallucination in Appendix L to further reinforce the practical value of our motivation (`Reviewer EXWb`)
- Provided detailed mathematical derivations with formal proofs in Appendix F to theoretically elaborate on how our method performs logits operations (`Reviewer nnUp`)
- Clarified NFA necessity beyond narrative purposes and explained our logic concept encompassing both question semantic logic and KG structural logic (`Reviewer EXWb`)
- Clarified MASK token handling and forward pass counting methodology to address `Reviewer nnUp`'s concerns about the methodological details of logits operations
- Discussed existing path-based approaches and KG-constrained generation methods in Appendix A to help readers understand the unique contributions and technical advantages of our work compared to existing input-level methods (`Reviewer EXWb, nnUp`)

---

> ### Author Response · Authors · 2025-12-02
> **Summary of Rebuttal Part 2**
>
> **2. Experimental Validation, Efficiency and Scalability**:
> - Supplemented comprehensive F1 score comparisons with recent baselines in Appendix C to demonstrate that our method not only achieves state-of-the-art performance in Hits@1 scores, but also attains advanced performance in F1 scores based on 13B models, while demonstrating that our approach exhibits more pronounced improvements on larger models (0.5B-13B range) (`Reviewer bu9C`)
> - Conducted experiments on additional model series including Microsoft, Mistral AI, and InternLM with larger parameter models up to 14B in Sec. 4.5 to comprehensively demonstrate the robustness and superior performance of Logits-to-Logic across different backbones (`Reviewer nv8o`)
> - Provided detailed computational overhead analysis showing minimal additional GPU usage and running time in Appendix I (`Reviewer EXWb, bu9C`)
> - Provided detailed computational overhead comparisons showing substantial advantages in LLM API calls and costs compared to existing agentic methods in Appendix I (`Reviewer EXWb, bu9C`)
> - Conducted comprehensive NFA construction overhead analysis demonstrating feasibility on large-scale KGs in Appendix J. The above supplemented efficiency analysis and experiments address the concerns of `Reviewers nnUp, EXWb, bu9C` regarding the computational overhead of our method
>
> **3. Importance of Core Modules**:
> - Demonstrated effectiveness through extensive ablation studies showing both Logits Strengthening and Filtering contribute significantly to performance (`Reviewer EXWb`)
> - Supplemented explanations of core differences between our approach and input-level methods like GCR through performance gains to illustrate our method's contribution in providing deeper insights into logic-consistent reasoning (`Reviewer bu9C`)
>
> **4. Evaluation Comprehensiveness and Fairness**:
> - Addressed dataset currency concerns by demonstrating evaluation covers the most extensive and commonly used KGQA benchmarks (`Reviewer nnUp`)
> - Clarified that our framework is training-free with only one hyperparameter $\omega$, and addressed `Reviewer nv8o`'s concerns regarding class imbalance by explaining our SFT is only for output format learning without introducing prior knowledge and unfair advantages (`Reviewer nnUp`)
>
> We are grateful for responses **from `Reviewers bu9C, EXWb, nnUp`, who acknowledged that their concerns have been resolved** (e.g., `Reviewer nnUp` noted: *Thank you for the detailed explanations and additional experiments. My concerns regarding W1, W2.3, W3, W4, Q1-4 are primarily resolved*; `Reviewer bu9C` noted *Thank you for the authors' feedback. I now have a better understanding of the paper*). We also actively addressed their follow-up questions and believe our detailed responses meet the expectations of resolving their concerns.
>
> We appreciate the AC's efforts and thank all reviewers for their dedicated feedback throughout this review process. Their insightful guidance has significantly enhanced our paper's quality and strengthened our confidence that this work will provide valuable contributions to the ICLR community and the broader field of structured knowledge reasoning.

---

### Author Response · Authors · 2025-11-24
**A Gentle Request for Your Participation in the Rebuttal Discussion**

Dear Reviewers,

Thank you for your thorough review and constructive feedback on our manuscript. We have carefully addressed all the points raised in your comments and have updated our paper accordingly (revisions highlighted in the revised manuscript).

We hope our responses and revisions have satisfactorily resolved the concerns you initially identified. As the discussion period nears its end, if our responses have addressed some of your concerns, we would be truly grateful if you could consider revising your rating score for our paper. Your support and consideration mean a great deal to us.

We sincerely appreciate, once again, your time and thoughtful feedback, and we look forward to your response at your earliest convenience.

Best regards, The Authors

---

> ### Author Response · Authors · 2025-11-28
>
> Dear Reviewers,
>
> I hope this message finds you well. As the discussion period is nearing its end with less than five days remaining, I wanted to ensure we have addressed all your concerns satisfactorily. Please do not hesitate to let us know if there is anything further we could do to improve your impression and final rating of our work. Your insights are invaluable to us, and we're eager to address any remaining issues to improve our work.
>
> Thank you for your time and effort in reviewing our paper.

---

### Meta-Review · Area_Chair_viSK · 2026-01-06

**Summary:**

This work initially received evenly mixed scores on either side of the border line. Post-rebuttal, likely some scores would be raised but some maintained, making it overall positive-leaning but still very much on the borderline.

More important, beyond numerical scores, while the rebuttals were thorough and the 3/4 reviewers responded, only 1 clearly stated their concerns were addressed, while 1 indicated they were not convinced. Crucially, there was only scant reviewer support or enthusiasiam about the novelty or significance of the work; the listed strengths were primarily about motivation, interesting problem and results.

Unfortunately, while overall there no major unaddressed technical concerns, the general lukewarm regard by reviewers weighs down the paper. For a highly-selective venue like ICLR, in comparison with other better-regarded submissions, I do not recommend acceptance this time round.

**Reviewer Concerns:**

Pls see above

**Reviewer Scores:**

-- nnUP and EXWb: based on their acknowledgement of the rebuttal (but without strong indication of raising scores), I would hazard 50/50 chance of raising their scores.
-- bu9C: based on their response to rebuttal, likely to maintain at 4.
-- Nv80: I would guess they'd maintain their positive-leaning score of 6.

---

### Decision · Program_Chairs · 2026-01-26

Reject